# Spatially resolved in silico modeling of NKG2D signaling kinetics suggests a key role of NKG2D and Vav1 Co-clustering in generating natural killer cell activation

Rajdeep Kaur Grewal[1], Jayajit Das[1,2,3,4,5,6]*

**1** Battelle Center for Mathematical Medicine, Abigail Wexner Research Institute, Nationwide Children's Hospital, Columbus, Ohio, United States of America, **2** Biomedical Sciences Graduate Program, The Ohio State University, Columbus, Ohio, United States of America, **3** Department of Pediatrics, The Ohio State University, Columbus, Ohio, United States of America, **4** Pelotonia Institute for Immuno-Oncology, The Ohio State University, Columbus, Ohio, United States of America, **5** Department of Biomedical Informatics, College of Medicine, The Ohio State University, Columbus, Ohio, United States of America, **6** Biophysics Graduate Program, The Ohio State University, Columbus, Ohio, United States of America

* jayajit@gmail.com

**Data Availability Statement:** Codes describing our in silico models and parameter estimation are available at the link https://github.com/jayajitdas/NK_signaling_spatial_model. All other relevant

## Abstract

Natural Killer (NK) cells provide key resistance against viral infections and tumors. A diverse set of activating and inhibitory NK cell receptors (NKRs) interact with cognate ligands presented by target host cells, where integration of dueling signals initiated by the ligand-NKR interactions determines NK cell activation or tolerance. Imaging experiments over decades have shown micron and sub-micron scale spatial clustering of activating and inhibitory NKRs. The mechanistic roles of these clusters in affecting downstream signaling and activation are often unclear. To this end, we developed a predictive in silico framework by combining spatially resolved mechanistic agent based modeling, published TIRF imaging data, and parameter estimation to determine mechanisms by which formation and spatial movements of activating NKG2D microclusters affect early time NKG2D signaling kinetics in a human cell line NKL. We show co-clustering of NKG2D and the guanosine nucleotide exchange factor Vav1 in NKG2D microclusters plays a dominant role over ligand (ULBP3) rebinding in increasing production of phospho-Vav1(pVav1), an activation marker of early NKG2D signaling. The in silico model successfully predicts several scenarios of inhibition of NKG2D signaling and time course of NKG2D spatial clustering over a short (~3 min) interval. Modeling shows the presence of a spatial positive feedback relating formation and centripetal movements of NKG2D microclusters, and pVav1 production offers flexibility towards suppression of activating signals by inhibitory KIR ligands organized in inhomogeneous spatial patterns (e.g., a ring). Our in silico framework marks a major improvement in developing spatiotemporal signaling models with quantitatively estimated model parameters using imaging data.

data are within the manuscript and its Supporting Information files.

**Funding:** This work was funded by the National Institutes of Health (NIH) (R01-AI 143740 and R01-AI 146581) and Nationwide Children's Hospital through grants to JD. The funders had no role in study design, data collection and analysis, decision to publish, or preparation of the manuscript.

**Competing interests:** The authors have declared that no competing interests exist.

## Author summary

Natural Killer cells are lymphocytes of our innate immunity and provide important resistance against viral infections and tumors. NK cells scan the local environment with diverse activating and inhibitory NK cell receptors (NKRs) and remain tolerized or lyse target cells expressing cognate ligands to NKRs. NKRs have been found to form micron sized clusters (or microclusters) as they interact with cognate ligands, and mechanisms regarding how the formation and movements of these microclusters influence NK cell signaling and activation, specifically related to activating NKRs, are often unclear. To this end, we develop a predictive spatially resolved early-time NK cell signaling model to study the interplay between membrane-proximal biochemical signaling events and the kinetics of microclusters of activating NKG2D and inhibitory KIR2DL2 receptors. We used published TIRF imaging data to validate our in silico models and estimate model parameters. Predictions from multiple in silico models are tested against a variety of data obtained from published imaging experiments and immunoassays. Our analysis suggests co-clustering of NKG2D and the guanosine nucleotide exchange factor Vav1 in the microclusters plays a major role in enhancing downstream activating signals. The developed framework can be extended to describe spatiotemporal signaling for other activating NKRs including CD16.

## Introduction

Natural Killer (NK) cells are lymphocytes of our innate immune system which provide important immune protection against viral infection and tumors [1,2]. NK cells express a wide range of germ line encoded activating and inhibitory NK cell receptors (NKRs). In humans, activating NKRs include NKG2D and killer Ig- like receptor (KIR) KIR2DS1, and inhibitory NKRs include inhibitory KIRs such as KIR2DL1, and KIR2DL2. NKRs bind to cognate ligands expressed by target cells initiating biochemical, physical, and mechanical modifications within NK cells that culminate into diverse NK cell responses ranging from a neutral response to lysis of target cells to secretion of cytokines [3]. Healthy host cells express polymorphic class I HLA molecules, cognate to a wide range of inhibitory NKRs, and generate tolerant NK cell responses. Whereas, tumor and infected cells downregulate expression of class I HLA molecules or upregulate expressions of ligands cognate to activating NKRs and tip the balance between activating and inhibitory signals toward NK cell activation. NKG2D is one of the best-studied activating NKR. In humans it binds to two families of ligands: one family (MICA and MICB) is akin to MHC class I and the other (ULBP1-3) is related to human proteins that bind to UL16 protein of human cytomegalovirus [4]. Tumor or infected host cells upregulate expressions of NKG2D ligands which contribute to lysis of these cells by NK cell cytotoxic responses [4].

Spatial clustering of NKG2D has been well investigated in confocal [5], total internal reflection fluorescence (TIRF) [6], and super-resolution microscopy experiments [7,8]. Abeyweera et al. [6] using TIRF microscopy experiments reported formation of mobile and immobile microclusters of NKG2D in human NK cell line NKL stimulated by cognate ligand ULBP3 presented on a planar lipid bilayer supported by a glass coverslip. The NKG2D microclusters form at the interface between the NK cell and the lipid bilayer which is also known as the immunological synapse (IS). NKG2D microclusters that are generated at the periphery of IS migrate to the central region of the IS at later times, whereas NKG2D microclusters formed at the central region of the IS remain immobile. The mobility of NKG2D microclusters depends

on actin remodeling as treatment by latrunculin, an actin depolymerizing agent, abrogate microcluster movements [6]. Additionally, confocal microscopy experiments show simultaneous localization of NKG2D receptors, Grb2, and Vav1 in the IS in NK cell line NKL upon stimulation by NKG2D ligand MICA [7]. Phosphorylation of Vav1 induces actin remodeling via activation of Rac GTPases [9,10], and thus can regulate motility of NKG2D microclusters.

Spatially resolved computational models have been successfully employed to glean mechanisms that underlie formation, motility, and function of spatial clusters of activating [11] and inhibitory [12,13] NKRs. Kaplan et al. [11] developed a spatially resolved model to investigate hypotheses regarding signal integration of activating NKG2D and inhibitory human KIR2DL/ mouse Ly49 receptors and concluded inhibitory NKRs locally suppress activating signals. Spatial in silico models describing clustering and signaling of inhibitory NKRs elucidated mechanisms giving rise to peptide antagonism for inhibitory KIR2DL2/3[12]. However, the above in silico models do not quantitatively match or fit microcluster formation and movements of NKRs with that observed in microscopy experiments. As we reason below, in silico modeling of spatial kinetics of NKG2D microclusters is important for gleaning mechanisms and generating improved model predictions. (1) Formation of NKG2D microclusters can increase the production of phosphorylated Vav1 (pVav1) due to increase in the frequency of ligand (e.g., ULPB3) rebinding to NKG2D residing within microclusters, and/or increase in biochemical propensity of signaling reactions when NKG2D molecules are co-clustered with other signaling molecules (e.g., Vav1). (2) Increased production of pVav1 due to clustering of NKG2D can increase centripetal movements of NKG2D microclusters leading to higher spatial aggregation of NKG2D thereby further increasing pVav1 production. This chain of events constitutes a "spatial" positive feedback [14]. Therefore, sizes and spatial separations between NKG2D microclusters could be relevant for affecting downstream signaling.

Motivated by above reasoning and a potential interplay between signaling kinetics and spatial clustering, we developed a framework combining a spatially resolved mechanistic agent based model and published TIRF imaging data assaying spatiotemporal signaling kinetics in NKL cell lines stimulated via NKG2D receptor or NKG2D and KIR2DL2 receptors. The agent based model developed here represents a major improvement over previous modeling efforts in the following aspects: (1) The model is able to quantitatively describe micron scale details of spatial clustering of activating and inhibitory NKRs. (2) A detailed parameter estimation of model parameters is carried out using spatial data. (3) The model includes interplay between spatial clustering and activating NKR signaling kinetics. We investigated three hypotheses (Model 1, Model 2, and Model 3) to show that co-clustering of Vav1 with NKG2D rather than ligand rebinding of ULBP3 is required to produce increased pVav1 production due to the formation of NKG2D microclusters. The presence of the spatial positive feedback allows for an efficient suppression of early time NKG2D signaling by heterogeneously distributed inhibitory HLA-C ligands on target cells.

## Model development

We developed a spatially resolved agent based model involving activating NKG2D receptors, inhibitory KIR2DL2 receptors, cognate NKR ligands (ULBP3 and HLA-C), and signaling proteins: Src family kinases (SFKs), Vav1, and phosphatase SHP1. The NKRs and signaling proteins interact via different rules to describe membrane proximal signaling events in human NK cell line NKL. The model includes rules describing biochemical signaling reactions, spatial movements of signaling complexes, and regulation of the spatial movements by biochemical reactions. The rules in the model are designed to quantitatively capture several distinguishing features of spatiotemporal NK cell signaling kinetics observed in previous imaging

experiments, namely, a) formation of mobile and immobile NKG2D microclusters upon NKG2D stimulation[6], b) centripetal movements of the mobile NKG2D microclusters from the periphery to the central region of the IS[6,15], c) decrease in the velocity of mobile NKG2D microclusters as those move closer to the central region of the IS[6], and, d) random movements of NKG2D microclusters in the central region of the IS[6].

Our agent based model describes spatiotemporal membrane proximal NK cell signaling in a quasi three-dimensional simulation box representing the interface between NK cell membrane and the plasma membrane of target cell or the lipid bilayer in a TIRF experiment at the IS (Fig 1). The simulation box of area (A = 15μm × 15μm) and thickness $z$ is discretized into chambers of volume $l_0 \times l_0 \times z$ where, $z = l_0$ or $z = 2l_0$ ($l_0 = 0.5$μm) for molecules residing in the plasma membrane or the cytosol, respectively. The following spatiotemporal processes occur in the models.

**(i) Biochemical signaling reactions.** We considered key membrane proximal biochemical signaling reactions involved in NKG2D and KIR2DL2 signaling. The specific roles of signaling proteins (adaptors, kinases, GEFs) considered here in regulating NKG2D signaling have been validated in experiments and modeling over the years. A selection of the pertaining literature is cited below and in Table 1. Similar to Mesecke et al. [16], we take a parsimonious approach in creating the NKG2D signaling network where key signaling reactions are

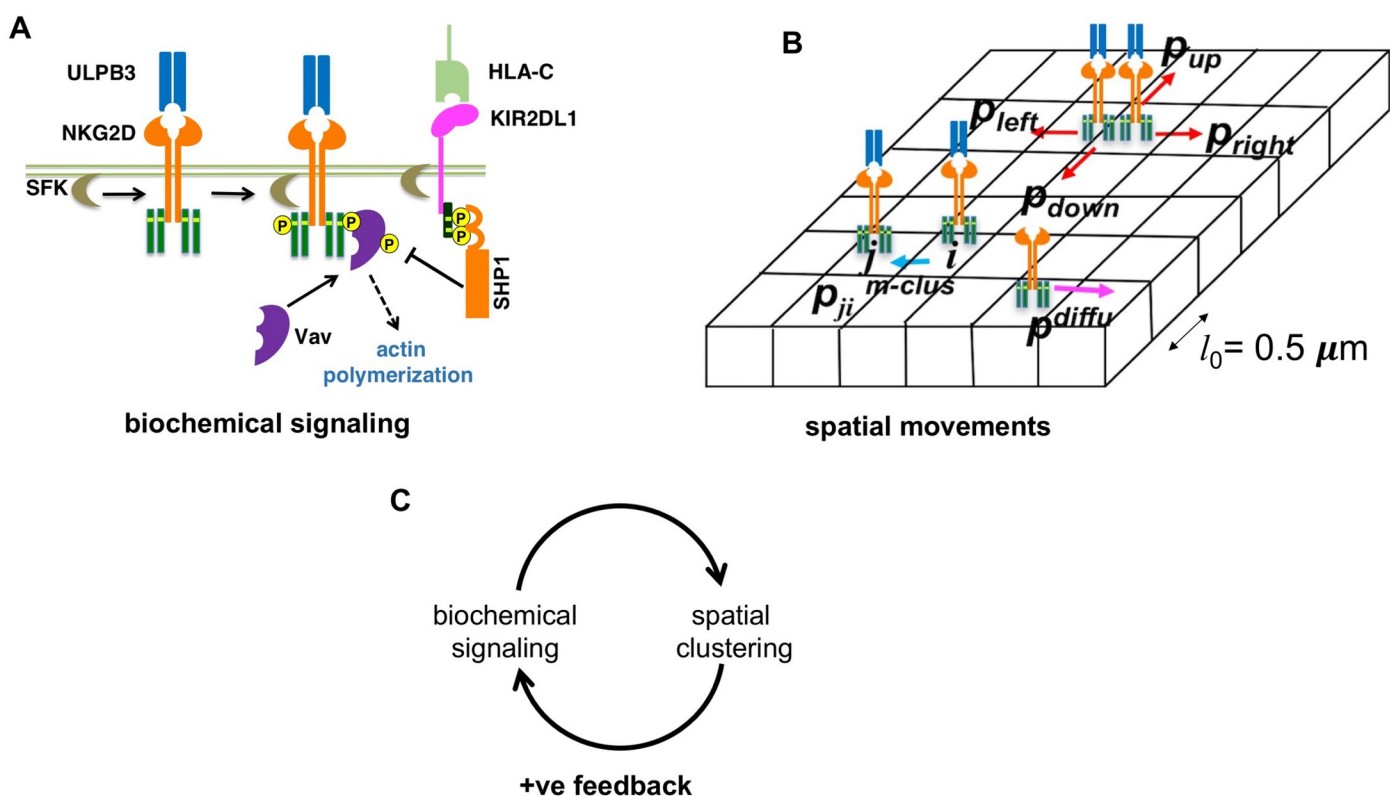

**Fig 1. Schematic depiction of the agent based models. (A)** Shows biochemical signaling reactions considered in the models. The reactions and their propensities are shown in Table 1. **(B)** Shows spatial movements considered in the agent based models. The simulation box is divided into chambers of volume $l_0 \times l_0 \times z$. A ULBP3 bound NKG2D complex at chamber $i$ moves to its nearest neighbor chamber $j$ with a probability $p^{(\text{m-clus})}_{ji}$, $p^{(\text{m-clus})}_{ji}$ depends on the number of pVav1 molecules in chamber $i$ and the four nearest neighbor chambers. ULBP3 bound NKG2D complexes in a chamber hop to the nearest neighbor chambers with probabilities $p_{\text{left}}$, $p_{\text{right}}$, $p_{\text{down}}$ and $p_{\text{up}}$ to implement centripetal movements (see main text). Free protein (receptors, kinases, phosphatases, Vav1/pVav1) molecules hop to next nearest neighbor chambers with probability $p^{\text{diffu}}$ implementing diffusive moves. (C) The biochemical signaling induces spatial clustering of NKG2D which in turn increases biochemical signaling- this represents a spatial positive feedback in the models.

considered. This approach is widely used for developing "detailed but manageable" signaling kinetic models where a model contains select reactions that pertain to the experimental data the model aims to describe [17]. Below we describe signaling reactions and specific approximations considered in the model.

NKG2D receptors bind to cognate ligands (ULBP3) to form NKG2D-ULPB3 complexes. NKG2D homodimers are associated with a pair of DAP10 homodimers in human NK cells. The tyrosine residues in two YINM motifs in a DAP10 homodimer are phosphorylated by Src family kinases [18,19] upon ULBP3 binding. NK cells contain several Src family kinases including Lck, Fyn, Lyn, and Yes [20]. During signaling the DAP10 molecules associated with NKG2D homodimers can be in a variety of partially phosphorylated states where one, two, or three of the total four tyrosine residues are phosphorylated. To reduce the number of agents in the model we approximated partially and fully phosphorylated states of the two DAP10 homodimers by two states, unphosphorylated or fully phosphorylated. In the model, a kinase molecule (SFK) represents multiple Src family kinases that phosphorylate DAP10 upon the formation of the NKG2D-ULBP3 complex. In NK cells, phosphorylated DAP10 becomes available to bind Grb2-Vav1 complex [1,16,21]. In the model, these reactions are approximated by binding of Vav1 to fully phosphorylated DAP10 where Grb2 is not included explicitly. Tyrosine residues in Vav1 have been found to be phosphorylated by SFKs in in vitro assays [22,23]. In NK cells, engagement of adhesion receptor LFA-1[24] or 2B4 receptors [25] has been reported to lead to phosphorylation of tyrosine residues in Vav1. In an experimentally validated in silico NKG2D signaling model Mesceke et al. [16] considered phosphorylation of tyrosine residues in Vav1 by SFKs. Given the above background we considered DAP10 bound Vav1 is phosphorylated by the SFK in the model. Vav1 phosphorylation is an important event during early time NK cell signaling as pVav1 leads to actin polymerization and degranulation in NK cells [2,26]. Inhibitory KIR2DL2 receptors bind to cognate ligands (HLA-C) and tyrosine residues in immunoreceptor tyrosine based inhibitory motifs (ITIMs) associated with the cytoplasmic part of KIR2DL2 are phosphorylated by the SFKs [27]. SFKs have been implicated in phosphorylation of tyrosine residues on ITIMs [27,28], however, the precise mechanisms are not clear [29]. Phosphorylation of tyrosine residues in ITIMs results in recruitment of SHP-1[29,30] which lead to dephosphorylation of pVav1[29,31]. We assume the catalytic domains of SFK phosphorylate the ITIMs associated with KIR2DL2. The model assumes two states of ITIM phosphorylation (unphosphorylated and fully phosphorylated) and the phosphatase SHP-1 binds to fully phosphorylated ITIMs. ITIM bound SHP-1 dephosphorylates pVav1 via enzymatic reactions. Unbinding of ligands from cognate receptors (NKG2D or KIR2DL2) dissociates the signaling complexes completely in the model–this step is included to implement kinetic proofreading [32,33]. In addition, there are first order dephosphorylation reactions for pVav1 representing dephosphorylation of phospho-tyrosines by phosphatases other than SHP1[34]. Some of the biochemical signaling reactions were not included in the model to keep the model simple and to stay focused on questions of interest, which is further discussed in the Limitations of the model section at the end.

**(ii) Spatial movements.** We modeled movements of NKG2D-ULBP3 complexes for formation of NKG2D microclusters and centripetal movements of NKG2D microclusters towards the center of the IS. These movements are assumed in the model to be dependent on actin remodeling which is regulated by signaling products such as pVav1[35]. The above movements are implemented by hops to nearest neighbor chambers occurring with specific probabilities. The velocity of the microclusters decreases as those move closer to the center. In addition, unbound molecules of receptor, ligand, and signaling proteins perform diffusive random movements in the model. Further details are provided in Table 1, Materials and Methods section, and the Supplementary Material.

**Table 1. List of processes and parameter values used in the agent based model.** $n_x$ is the number of species $x$ in a chamber. $k^{(x\text{-}y)}_{on}$ denotes binding rate for species $x$ and $y$. $k^{(x\text{-}y)}_{off}$ denotes unbinding rate of complex $x$-$y$. $k^{(x\text{-}E/P)}_{phospho/dephospho}$ is the catalytic rate of phosphorylation/de-phosphorylation of $x$ by enzyme E/P. $k^{(Px)}_{dephospho}$ denotes the de-phosphorylation rate of $x$ by uncharacterized phosphatases.

| Rule | Processes implemented in the model | Propensities | Notes | Range Used in PSO |
|---|---|---|---|---|
| | | Biochemical signaling processes initiated by NKG2D | | |
| 1. | NKG2D receptors binding/ unbinding with ligand (ULBP3) | Binding: $k^{(NKG2D\text{-}ULBP3)}_{on}$ $n_{NKG2D}$ $\times$ $n_{ULBP3}$ | $k_{on}$ estimated by PSO. Range fixed by using $K_D$ = 4μM for ULBP3 [36] and $k_{off}$ = 0.023s⁻¹ (for MICA)[36] | $k^{(NKG2D\text{-}ULBP3)}_{on}$: $6\times10^{-5}$ $- 6 \times10^{-2}$ μM⁻¹s⁻¹ |
| | | Unbinding: $k^{(NKG2D\text{-}ULBP3)}_{off}$ $n_{NKG2D\text{-}ULBP3}$ | $k_{off}$ fixed to 0.023s⁻¹, also close to the measured $k_{off}$ for ULPB1[37]. | |
| 2. | Binding/unbinding of SFKs to DAP10 | Binding: $k^{(DAP10\text{-}SFK)}_{on}$ $n_{DAP10}$ $\times$ $n_{SFK}$ *$n_{DAP10}$ is equal to $n_{NKG2D}$ in the model. | Parameters estimated by PSO; ranges are based on our estimation lck catalytic domain:tyrosine (on CD3ζ) binding reaction rate $k_{on}$ = 0.063 $(\mu m)^2 s^{-1}$ using Ref. [38]. Further details in S1 Text and S14 Fig. | $k^{(DAP10\text{-}SFK)}_{on}$: 1.1–1.1× 10² μM⁻¹s⁻¹ |
| | | Unbinding: $k^{(DAP10\text{-}SFK)}_{off}$ $n_{DAP10\text{-}SFK}$ *$n_{DAP10}$ is equal to $n_{NKG2D}$ in the model. | Parameter estimated by PSO; based on the catalytic- domain of Csk:tyrosine (on Lck) dissociation rate, $k_{off}$ = 0.044 s⁻¹ [39]. Further details in S1 Text and S14 Fig. | $k^{(DAP10\text{-}SFK)}_{off}$: 0.006–6 s⁻¹ |
| 3. | Phosphorylation of tyrosine residues in adaptor DAP10 via SFKs | Phosphorylation: $k^{(DAP10\text{-}SFK)}_{phospho}$ $n_{DAP10\text{-}SFK}$ | Parameter estimated by PSO; based on ITAM-tyrosine:lck phosphorylation kinetics, $k_{cat}$ = 2.3×10⁻⁴ to 98.4 × 10⁻⁴ s⁻¹ [40]. | $k^{(DAP10\text{-}SFK)}_{phospho}$: 0.01–10 s⁻¹ Upper limit is chosen to be 1000 × the reported value to match the kinetics at seconds scale in the simulations. |
| 4. | Vav1 binding/unbinding to phosphorylated DAP10 | Binding: $k^{(pDAP10\text{-}Vav1)}_{on}$ $n_{pDAP10}$ $\times$ $n_{Vav1}$ | Parameter estimated by PSO; based on $K_D$ = 16.8μM for Vav-SH3:Grb2-SH3 binding kinetics[41]. and $k_{off}$ = 10 s⁻¹ (0.01 s⁻¹) which gives $k_{on}$ = 6 ×10⁻¹ μM⁻¹s⁻¹ (6 × 10⁻⁴ μM⁻¹s⁻¹) | $k^{(pDAP10\text{-}Vav1)}_{on}$: 0.6–60.0μM⁻¹s⁻¹ Due to small binding rate the upper limit is chosen to be 100 times the reported value |
| | | Unbinding: $k^{(pDAP10\text{-}Vav1)}_{off}$ $n_{pDAP10\text{-}Vav1}$ | Assumed to lie between 0.01–10 s⁻¹ (Range similar to reactions considered here). | $k^{(pDAP10\text{-}Vav1)}_{off}$: 0.01–10 s⁻¹ |
| 5. | Phosphorylation of Vav1 by SFKs | Phosphorylation: $k^{(pDAP10\text{-}Vav1\text{-}SFK)}_{phospho}$ $n_{pDAP10\text{-}Vav1\text{-}SFK}$ | Parameter estimated by PSO; based on Vav1-tyrosine:lck-SH2 phosphorylation kinetics $k_{cat}$ = 1.78 s⁻¹ to 2.4 s⁻¹ [42]. | $k^{(pDAP10\text{-}Vav1\text{-}SFK)}_{phospho}$: 0.01–10 s⁻¹ |
| 6. | Binding/unbinding of SFKs to pDAP10-Vav1 | Binding: $k^{(pDAP10\text{-}Vav1\text{-}SFK)}_{on}$ $n_{pDAP10\text{-}Vav1}$ $\times$ $n_{SFK}$ | Parameter estimated by PSO; based on Vav1-phospho-tyrosine:lck-SH2 phosphorylation kinetics $K_D$ = 1.39μM [42] and $k_{off}$ = 10 s⁻¹ (0.01 s⁻¹) which gives $k_{on}$ = 7.2 μM⁻¹s⁻¹ (7 x 10⁻³ μM⁻¹s⁻¹) | $k^{(pDAP10\text{-}Vav1\text{-}SFK)}_{on}$: $7.6 \times 10^{-3}$–7.6 μM⁻¹s⁻¹ |
| | | Unbinding: $k^{(pDAP10\text{-}Vav1\text{-}SFK)}_{off}$ $n_{DAP10\text{-}Vav1\text{-}SFK}$ | Assumed to lie between 0.01–10 s⁻¹ (range similar to reactions considered here) | $k^{(pDAP10\text{-}Vav1\text{-}SFK)}_{off}$: 0.01–10 s⁻¹ |
| 7. | De-phosphorylation of NKG2D-pDAP10 by phosphatases | De-phosphorylation: $k^{(pDAP10)}_{dephospho}$ $n_{pDAP10}$ | Parameter estimated by PSO; based on dephosphorylation of NKG2D-pDAP10, $k_{cat}$ = 0.245 s⁻¹ [43] | $k^{(pDAP10)}_{dephospho}$: 0.02–2 s⁻¹ |
| 8. | De-phosphorylation of pVav1 by phosphatases | De-phosphorylation: $k^{(pDAP10\text{-}pVav1)}_{dephospho}$ $n_{pDAP10\text{-}pVav1}$ | Parameter estimated by PSO; based on dephosphorylation of pVav1, $k_{cat}$ = 0.145 s⁻¹ [43] | $k^{(pDAP10\text{-}pVav1)}_{dephospho}$: 0.01–1 s⁻¹ |
| | | Biochemical signaling processes initiated by inhibitory KIR2DL2 | | |
| 9. | KIR2DL2 receptors binding/ unbinding with ligand (HLA-C) | Binding: $k_{on}^{(KIR2DL2)}$ $n_{KIR2DL2}$ $\times$ $n_{HLA\text{-}C}$ | Fixed value. Based on KIR2DL2:HLA-C binding kinetics with $K_D$ = 3.6x10⁻² μM [44] and $k_{off}$ = 1 s⁻¹ which gives $k_{on}$ = 27.7 μM⁻¹ s⁻¹ | $k_{on}^{(KIR2DL2)}$: 27.7 μM⁻¹ s⁻¹ |
| | | Unbinding: $k_{off}^{(KIR2DL2)}$ $n_{KIR2DL2\text{-}HLA\text{-}C}$ | Fixed value. Based on KIR2DL3:HLA-Cw7 binding kinetics with $k_{off}$ = 1.1 s⁻¹ [45] | $k_{off}^{(KIR2DL2)}$: 1.0 s⁻¹ |
| 10. | Binding/unbinding of SFKs to ITIMs in KIR2DL2 | Binding: $k^{(ITIM\text{-}SFK)}_{on}$ $n_{ITIM}$ $\times$ $n_{SFK}$ * $n_{ITIM}$ = $n_{KIR2DL2}$ in the model. | Fixed value. Assumed to be same as rule #2; estimated from PSO. | |
| | | Unbinding: $k^{(ITIM\text{-}SFK)}_{off}$ $n_{ITIM\text{-}SFK}$ * $n_{ITIM}$ = $n_{KIR2DL2}$ in the model. | Fixed value. Assumed to be same as rule #2; estimated from PSO. | |

*(Continued)*

**Table 1.** (Continued)

| Rule | Processes implemented in the model | Propensities | Notes | Range Used in PSO |
|------|-----------------------------------|--------------|-------|-------------------|
| 11. | Phosphorylation of tyrosine residues in ITIMs via SFKs | Phosphorylation: $k^{(ITIM-SFK)}_{phospho}$ $n_{ITIM-SFK}$ | Fixed value. Assumed to be same as rule #3; estimated from PSO. | |
| 12. | SHP1 binding/unbinding to phosphorylated ITIM | Binding: $k^{(pITIM-SHP1)}_{on}$ $n_{pITIM} \times$ $n_{SHP1}$ | Fixed value. Based on SHP1-Ly49A binding kinetics with $k_{on} = 37.40 \times 10^4$ $M^{-1}s^{-1}$ [46] | $k^{(pITIM-SHP1)}_{on}$: $36.1 \times 10^{-2}$ $\mu M^{-1}s^{-1}$ |
| | | Unbinding: $k^{(pITIM-SFK)}_{off}$ $n_{pITIM-SHP1}$ | Fixed value. Based on SHP1-Ly49A binding kinetics with $k_{off} = 5.09 \times 10^{-4}$ $s^{-1}$ [46] | $k^{(pITIM-SFK)}_{off}$: $5.09 \times 10^{-4}$ $s^{-1}$ |
| | Biochemical processes for inhibition of activating signaling by inhibitory signaling | | | |
| 13. | Binding/unbinding of ITIM bound SHP1 to pVav1 | Binding: $k^{(SHP1-pVav1)}_{on}$ $n_{bound\text{-}SHP1}$ $\times$ $n_{pVav1}$ | Fixed value. Assumed to be 6.11 $\mu M^{-1}$ $s^{-1}$. Further details are provided in S2 Text and S15 Fig. | $k^{(SHP1-pVav1)}_{on}$: 6.11 $\mu M^{-1}$ $s^{-1}$ |
| | | Unbinding: $k^{(SHP1-pVav1)}_{off}$ $n_{bound\text{-}SHP1-pVav}$ | Fixed value. Assumed to be 0.01 $s^{-1}$. Further details are provided in S2 Text and S15 Fig. | $k^{(SHP1-pVav1)}_{off}$: 0.01 $s^{-1}$ |
| 14. | Dephosphorylation of pVav1 by ITIM bound SHP1 | Dephosphorylation: $k^{(SHP1-pVav1)}_{dephos}$ $n_{bound\text{-}SHP1-pVav1}$ | Fixed value. Based on dephosphorylation of PD1 tethered SHP1 molecules [47,48], we fixed $k_{cat} = 3.0$ $s^{-1}$. Further details in S2 Text and S15 Fig. | $k^{(SHP1-pVav1)}_{dephos}$: 3.0 $s^{-1}$ |
| | Microcluster formation and spatial movements | | | |
| 15. | NKG2D microcluster formation | $\min\left(\frac{\exp\left(\beta\left(E_i-E_j\right)\right)}{1+\exp\left(\beta\left(E_i-E_j\right)\right)}, 1\right) \times n_{NKG2D\_sp}$ | $\beta$ estimated by PSO. | $\beta$: 0.01–0.1 |
| 16. | NKG2D microcluster movements | $k^{(cluster-move)} \times p_{left/right/up/down}$ $p_{left/right/up/down}$ is described in Eq 1 and depends on parameters $w$ and $K$. | $k^{(cluster-move)}$, $w$, and $K$ are estimated via PSO. | $k^{(cluster-move)}$: 0.1–5 $K$: 102–1581 $w$: 0.79–1 $w$ is chosen closer to 1 favoring centripetal over random movements. |
| 17. | Influx of NKG2D | $k^{(influx)} \times n_{NKG2D}$ | $k^{(influx)}$ estimated by PSO. This rule is added to describe the influx of NKG2D observed in the TIRF experiments in Ref.[6]. | $k^{(influx)}$: $1 \times 10^{-5}$–$1 \times 10^{-2}$ $s^{-1}$ |
| 18. | Diffusion of membrane bound free NKG2D, KIR2DL1, cognate ligands | $D/l^2_0$ $n_{NKG2D/KIR2DL1}$ | D fixed to 0.01 $\mu m^2$ $s^{-1}$ | |
| 19. | Diffusion of cytosolic signaling proteins | $D/l^2_0$ $n_{Vav1/pVav1/SHP1}$ | D fixed to 10 $\mu m^2$ $s^{-1}$ | |
| | Protein concentrations | | | |
| | NKG2D | | Set to 8 molecules/$(\mu m)^2$ based on the estimated number of NKG2D in a single NKL cell of diameter 18$\mu m$[16]. | |
| | Lck | | Set to 698 molecules/$(\mu m)^2$ based on the estimated number of NKG2D in a single NKL cell of diameter 18$\mu m$ [16]. | |
| | Vav1 | | Set to 114 molecules/$(\mu m)^3$ based on the estimated number of NKG2D in a single NKL cell of diameter 18$\mu m$ [16]. | |
| | SHP1 | | Set to 2090 molecules/$(\mu m)^3$ based on the estimated number of NKG2D in a single NKL cell of diameter 18$\mu m$[16]. | |
| | KIR2DL2 | | Set to 106 molecules/$(\mu m)^2$ [49]. | |
| | ULBP3 | | Parameter estimated by PSO. | 1075–3468 molecules in the simulation box. |
| | HLA-C | | Set to 98 molecules/$(\mu m)^2$ [50]. | |

**(iii) Interplay between spatial movements and signaling reactions.** The probabilities of hops generating NKG2D microclusters and centripetal movements of NKG2D micro-clusters depend on the local and the total number of pVav1 in the simulation box,

respectively. The rules capture the interplay between biochemical signaling reactions and regulation of specific spatial movements by signaling reactions which in turn affects signaling reactions. For NKG2D signaling the above interplay represents a positive feedback [14].

**Hypotheses considered.** We considered three hypotheses encoded in models Model 1, Model 2, and Model 3. The models probe functional outcomes of different types of NKG2D clustering and interplay between NKG2D microclusters and biochemical signaling reactions. In Model 1, mobile NKG2D microclusters and NKG2D bound signaling protein molecules move toward the central region of the IS. In Model 2, NKG2D complexes and membrane proximal Vav1 molecules move simultaneously toward the central region of the IS. This Vav1 species in the model could potentially represent Vav1 molecules that are recruited to the plasma membrane via SOS1[51]. Because of the above rule, there is a strong co-clustering of NKG2D and Vav1 in Model 2. In Model 3, we studied outcomes of the absence of the positive feedback between NKG2D microcluster movement and signaling reactions. In Model 3, the rate of centripetal movements of mobile NKG2D microclusters is independent of pVav1 abundances. Model 3 contains co-clustering of NKG2D and Vav1 as in Model 2. The models are summarized in Table 2.

## Model simulation and parameter estimation

The simulations are carried out in a quasi-three-dimensional simulation box representing a thin junction between NK cell and the supported lipid bilayer in TIRF experiments. The simulation box has an area of $15 \times 15 \ \mu m^2$ and a depth of $z$, and is divided into small cubic chambers of size $(l_0 \times l_0 \times z)$, where $l_0 = 0.5 \mu m$ and $z = l_0$ (for molecules residing in membrane) or $2l_0$ (for cytosolic molecules). The molecules are well-mixed in each chamber and molecules in a chamber hop to next nearest neighbor chambers with specific rates to produce diffusive or centripetal movements, or movements leading to microcluster formation. The kinetics of the system is simulated using a kinetic Monte Carlo (kMC) approach via a freely available simulator SPPARKS (https://spparks.sandia.gov/). The kMC simulation includes intrinsic noise fluctuations in biochemical reactions as well as in spatial movements. The list of the processes, and their propensities are listed in Table 1. The copy numbers of most of the signaling proteins in the simulation box are estimated using available measured concentrations for NKLs (Table 1). However, values of many of the model parameters in the cellular environment are unknown, and we estimated these parameters by a parameter fitting scheme that reproduced the spatial pattern of NKG2D receptors measured in TIRF experiments [6]. The spatial patterns of NKG2D in our simulation and TIRF imaging data are quantified using mean values, variances, and a two-point correlation function computed from density profiles for NKG2D. Similar variables are widely used in statistical physics [52] to quantify spatial patterns. The Euclidean distance between dimensionless forms of the above variables in TIRF images and the agent based model is used to create a cost function which is minimized by particle swarm optimization (PSO) to estimate model

**Table 2. List of the agent based models.**

| | Biochemical signaling reactions | Formation of NKG2D microclusters | Centripetal movement of NKG2D microclusters | Co-clustering of NKG2D and Vav1 | pVav1 dependent centripetal movements of NKG2D micro clusters |
|---|---|---|---|---|---|
| **Model 1** | Yes | Yes | Yes | No | Yes |
| **Model 2** | Yes | Yes | Yes | Yes | Yes |
| **Model 3** | Yes | Yes | Yes | Yes | No |

parameters. Details regarding our parameter estimation scheme are provided in Materials and Methods section and the Supplementary Material.

# Results

## 1. Multiple models quantitatively describe kinetics of NKG2D microclusters

Spatiotemporal signaling kinetics of NKG2D and associated signaling proteins was simulated in Model 1 and Model 2. NKG2D, ULBP3, SFK, and Vav1 molecules were distributed homogeneously in space in the simulation box at the beginning of the simulation at t = 0. Binding of NKG2D and ULBP3 initiates a series of biochemical signaling reactions (Table 1) leading to phosphorylation of Vav1 in the simulations. The production of pVav1 molecules initiates NKG2D microcluster formation and centripetal movements of the NKG2D microclusters (S1 and S2 Movies). Both models fit spatial distribution of NKG2D in TIRF experiments at t = 1min, in particular, at length scales $\geq$ 1μm reasonably well (Fig 2). The variances and the two-point correlation functions calculated using spatial distributions of NKG2D in model simulations agreed well with that calculated from the TIRF imaging data (Figs 2 and S1). The estimated best-fit parameters for the models are shown in Table 3. Many parameter values show order of magnitude differences between Model 1 and Model 2, e.g., binding-unbinding rates

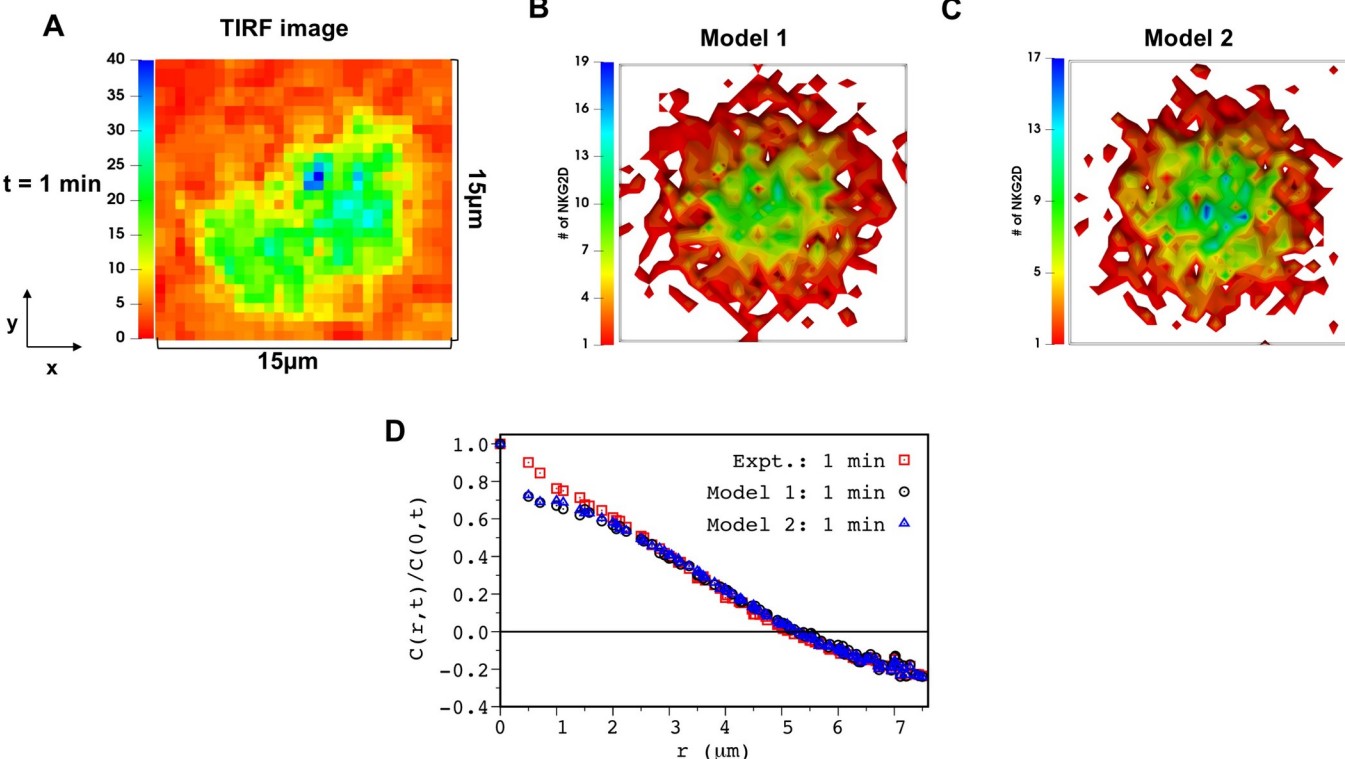

**Fig 2. Model 1 and Model 2 captures the spatial clustering of NKG2D (DAP10) in TIRF experiments. (A)** Shows a 2D fluorescence intensity of NKG2D-DAP10-mCherry in TIRF experiments in Ref. [6] at t = 1min post stimulation by ULPB3. The above image shows a region of interest extracted from S4 Fig in Ref. [6] which is coarse-grained to match the minimum length scale (~ 0.5 μm) of spatial resolution in our model (see Materials and Methods for details). **(B and C)** Shows spatial distribution of NKG2D in the x-y plane in the simulation box for our agent based model (B) Model 1 and (C) Model 2 at t = 1 min post NKG2D stimulation. The parameters for the simulation are set at the best fit value from our PSO. The area of the region of interest in the image and simulation box is set to 15μm × 15μm. **(D)** Shows comparison between the two-point correlation function (C(r,t)/C(0,t) vs r) at t = 1 min calculated from the image in (A) (red, empty square) and configurations for Model 1 (black, empty circle) and Model 2 (blue, empty triangle) shown in (B) and (C).

**Table 3. List of parameter values estimated from PSO.** Values within () and [] correspond to the standard deviation (s.d.) (details in Materials and Methods section) and to the ratio PSO s.d./(PSO estimated value), respectively.

| Processes | Parameters | PSO estimated parameters | |
|---|---|---|---|
| | | Model 1 | Model 2 |
| NKG2D receptors binding with ligand (ULBP3) | $k^{(NKG2D\text{-}ULBP3)}_{on}$ | $5.631 \times 10^{-2}\ \mu M^{-1}\ s^{-1}$ $(1.37 \times 10^{-2})$ [0.24] | $2.387 \times 10^{-2}\ \mu M^{-1}\ s^{-1}$ $(1.43 \times 10^{-2})$ [0.6] |
| Binding/unbinding of SFKs to DAP10 | $k^{(DAP10\text{-}SFK)}_{on}$ | $47.14\ \mu M^{-1}\ s^{-1}$ $(26.07)$ [0.55] | $1.054\ \mu M^{-1}\ s^{-1}$ $(8.56)$ [8.12] |
| | $k^{(DAP10\text{-}SFK)}_{off}$ | $1.831\ s^{-1}$ $(1.71)$ [0.93] | $0.006\ s^{-1}$ $(0.36)$ [61.29] |
| Phosphorylation of tyrosine residues in adaptor DAP10 via SFKs | $k^{(DAP10\text{-}SFK)}_{phospho}$ | $0.452\ s^{-1}$ $(1.89)$ [4.16] | $2.189\ s^{-1}$ $(1.94)$ [0.89] |
| Vav1 binding/unbinding to phosphorylated DAP10 | $k^{(pDAP10\text{-}Vav1)}_{on}$ | $35.4\ \mu M^{-1}\ s^{-1}$ $(10.0)$ [0.28] | $0.6112\ \mu M^{-1}\ s^{-1}$ $(3.64)$ [5.94] |
| | $k^{(pDAP10\text{-}Vav1)}_{off}$ | $5.069\ s^{-1}$ $(2.33)$ [0.46] | $0.01\ s^{-1}$ $(0.26)$ [26.16] |
| Binding/unbinding of SFKs to pDAP10-Vav1 | $k^{(pDAP10\text{-}Vav1\text{-}SFK)}_{on}$ | $1.023 \times 10^{-2}\ \mu M^{-1}\ s^{-1}$ $(0.38)$ [37.38] | $7.528\ \mu M^{-1}\ s^{-1}$ $(0.56)$ [0.07] |
| | $k^{(pDAP10\text{-}Vav1\text{-}SFK)}_{off}$ | $0.042\ s^{-1}$ $(1.13)$ [26.88] | $0.028\ s^{-1}$ $(0.14)$ [5.0] |
| Phosphorylation of Vav1 by SFKs | $k^{(pDAP10\text{-}Vav1\text{-}SFK)}_{phospho}$ | $0.922\ s^{-1}$ $(1.65)$ [1.79] | $0.776\ s^{-1}$ $(0.28)$ [0.37] |
| De-phosphorylation of pDAP10 by phosphatases | $k^{(pDAP10)}_{dephospho}$ | $0.127\ s^{-1}$ $(0.39)$ [3.03] | $2.0\ s^{-1}$ $(0.25)$ [0.13] |
| De-phosphorylation of pVav1 by phosphatases | $k^{(pDAP10\text{-}pVav1)}_{dephospho}$ | $0.048\ s^{-1}$ $(0.17)$ [3.63] | $1.0\ s^{-1}$ $(0.12)$ [0.12] |
| NKG2D microcluster formation | $\beta$ | $0.013$ $(8.43 \times 10^{-3})$ [0.64] | $0.01$ $(3.9 \times 10^{-4})$ [0.04] |
| NKG2D microcluster movements | $k^{(cluster\text{-}move)}$ | $0.899$ $(0.32)$ [0.35] | $1.157$ $(0.3)$ [0.26] |
| | $K$ | $102.086$ $(31.01)$ [0.30] | $102.086$ $(19.02)$ [0.19] |
| | $w$ | $0.802$ $(3.02 \times 10^{-2})$ [0.04] | $1.0$ $(5.37 \times 10^{-2})$ [0.05] |
| Influx of NKG2D | $k^{(influx)}$ | $6.6 \times 10^{-5}\ s^{-1}$ $(3.23 \times 10^{-4})$ [4.86] | $0.0044\ s^{-1}$ $(1.97 \times 10^{-3})$ [0.44] |
| Number of ULBP3 in the simulation box + outer rim | | $3459$ $(59.20)$ $[1.71 \times 10^{-2}]$ | $3462$ $(62.53)$ $[1.81 \times 10^{-2}]$ |

of SFK to DAP10 or of Vav1 to pDAP10. The reason for this difference can be explained as follows. In Model 2, co-clustering of Vav1 and NKG2D increases propensities of reactions that

influence pVav1 production and consequently the formation and motility of NKG2D microclusters. Therefore, different values for reaction rates (e.g., binding/unbinding rate of Vav1 to pDAP10) regulating microcluster kinetics are chosen as optimal parameter values in Model 1 to compensate for the absence of the increase in reaction propensities due to Vav1-NKG2D co-clustering in the model.

The computation of uncertainties in the estimated parameter values showed that about 2/3 of the total number of parameters are estimated well, e.g., (standard deviation)/(estimated value) < 3. The procedure for estimation of uncertainty is described in detail in the Materials and Methods section, and potential causes behind poor parameter estimations of 1/3 of the parameters are discussed in the Discussion section. Next, we assessed the utility of the parameter estimation scheme for generating predictions at future time points (Figs 3 and S2) and providing mechanistic insights (next section). The best-fit models were used for predicting spatial patterns of NKG2D at later times t > 1 min. Both models predict spatial distribution of NKG2D in TIRF imaging data until t = 3 min reasonably well (Figs 3 and S2). However, model predictions deviate from the C(r,t) data calculated for the TIRF images at t = 4 min (S3 Fig). This disagreement is likely due to change in the organization of the NKG2D microclusters caused by the spreading of the NK cells on the supported lipid bilayer which is not included in our agent based models.

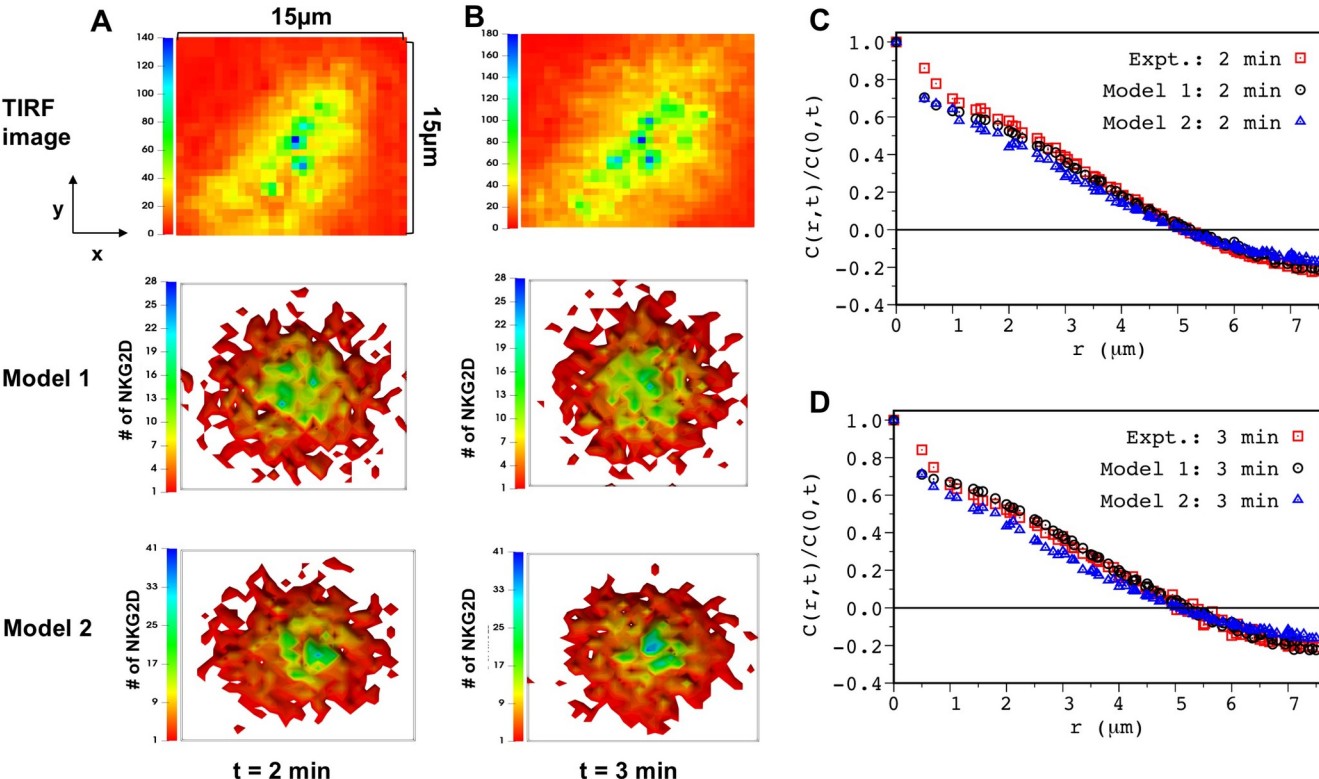

**Fig 3. Model prediction for NKG2D spatial clustering at later times are in agreement with TIRF experiments. (A)** (Top) Shows coarse-grained 2D image extracted for a region of interest in TIRF image (S4 Fig in Ref. [6]) of NKG2D-Dap10-mCherry at t = 2 min post stimulation by ULPB3. (Middle and Bottom) Shows spatial distribution of NKG2D in the x-y plane at t = 2 min post stimulation for Model 1 (middle) and Model 2 (bottom). The model parameters are set to the best fit values obtained for the fit at t = 1 min. **(B)** (top) Similar to (A), TIRF image Ref. [6] at 3 min post ULBP3 stimulation. (Middle and Bottom) Shows spatial distribution of NKG2D in the x-y plane at t = 3 min post stimulation for Model 1 (middle) and Model 2 (bottom). The model parameters are set to the best fit values obtained for the fit at t = 1 min. (C) Comparison of the two-point correlation function (C(r,t)/C(0,t) vs r) evaluated from the TIRF image (red, empty square) and simulations for Model 1 (black circle) and Model 2 (blue triangle) at t = 2 min. (D) Similar to (C) showing comparison between experiments and Model 1 and Model 2 at t = 3 min.

## 2. Co-clustering of NKG2D and Vav1 is required to increase the production of pVav1 due to the formation of NKG2D microclusters

We investigated the mechanistic role of formation and centripetal movement of NKG2D microclusters in increasing pVav1 production. The average lifetime of a NKG2D-ULBP3 complex within NKG2D microclusters can increase because of the increase in the frequency of ULBP3 rebinding due to higher density of NKG2D molecules in microclusters. The increased ULBP3 rebinding could elevate abundances of NKG2D-ULBP3 complexes leading to higher pVav1 production in Model 1 and Model 2. In addition, in Model 2, the increase in reaction propensities due to co-clustering of NKG2D and Vav1 can enhance pVav1 production. In order to evaluate the roles of ULBP3 rebinding and NKG2D and Vav1 co-clustering in increasing pVav1 production we compared kinetics of pVav1 production in Model 1 and Model 2 under two conditions: Case A: NKG2D is not allowed to form microclusters or perform centripetal movements, and, Vav1 does not co-cluster with NKG2D. This case represents NKG2D stimulation in experiments where NKG2D microcluster formation and migration is blocked by application of drugs inhibiting actin polymerization [14]. Case B: Spatial aggregations of NKG2D and Vav1 occur according to the model rules. Simulations for Case A result in spatially homogeneous distribution of NKG2D molecules in both models. Our simulations for Model 1 show that abundances of pVav1 at t = 1 min for a range of ULBP3 doses have negligible differences between Case A and B (Fig 4A). The kinetics of pVav1 production for a particular ULBP3 dose also shows almost no difference between Case A and B (S4A Fig). In contrast, pVav1 abundances decrease in Case A compared to Case B in Model 2 for a range of ULBP3 doses (Figs 4B and S4B). The magnitude of this decrease increases with the increasing ULBP3 dose (Fig 4B). These results suggest that co-clustering of NKG2D and Vav1 in Model 2 could be important for increased pVav1 production in the model.

However, ULBP3 rebinding could also help in elevating pVav1 production in Model 2. Therefore, we further quantified the contribution ULBP3 rebinding in increasing the average number of NKG2D-ULBP3 complexes in Model 2. We followed an approach in ref. [53] for this quantification, wherein decay kinetics of an initial fixed number of receptor-ligand complex is studied in the presence of immobile spatially distributed receptors. The simulations start with a fraction of receptors bound to ligands and no free ligands; free ligands created by dissociation of the receptor-ligand complex at a rate $k_{off}$ can diffuse and rebind to the receptors. In the absence of any ligand rebinding, the number of receptor-ligand complex decays exponentially as $\propto exp(-k_{off} t)$. The presence of rebinding produces a slower and a non-exponential decay of the number of receptor ligand complex with higher numbers of receptor-ligand complex remaining in the system at longer times ($t \gg 1/k_{off}$) compared to an exponential decay as $exp(-k_{off} t)$. We evaluated the decay kinetics of an initial number of NKG2D-ULBP3 complex where NKG2D-ULBP3 binding-unbinding reactions were the only reactions present in simulations. The NKG2D molecules, held fixed in space, were distributed uniformly and randomly or in a spatially clustered configuration obtained from our simulations for Model 1 or Model 2 at t = 1 min (Fig 2C). The decay kinetics shows non-exponential decay for both the cases and a small increase in the number of NKG2D-ULBP3 complex (Fig 4C and 4D) when NKG2D are clustered. The above results reveal that the effect of ULBP3 rebinding in our models is almost equally strong when NKG2D are distributed uniformly randomly or in microclusters. Thus, increased ULBP3 rebinding in NKG2D microclusters does not play a substantial role in increasing pVav1 production.

The increase in the production of pVav1 induced by NKG2D microclusters in Model 2 (Fig 4B) is consistent with experiments by Endt et al. [14], where blocking cytoskeletal

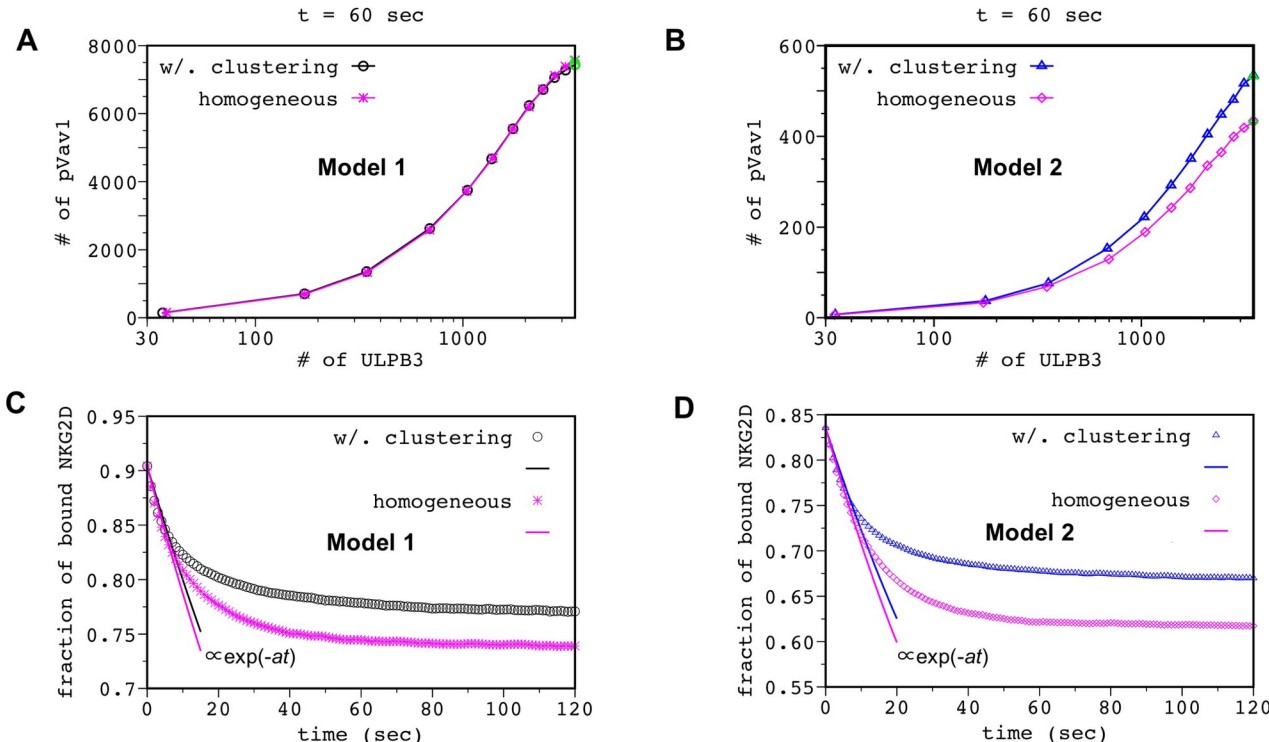

**Fig 4. Co-clustering of Vav1 is needed to increase pVav1 upon NKG2D stimulation.** Shows increase in the total number of pVav1 at t = 1 min as the number of ULBP3 is increased in Model 1 (**A**) and Model 2 (**B**) for cases where NKG2D are not allowed to form microclusters (magenta asterisks, Model 1; magenta diamonds, Model 2) or form microclusters (black circles, Model 1; blue triangles, Model 2) according to the model rules. The pVav1 concentrations are averaged over 50 different configurations. The points in green indicate the data obtained for parameter values that best-fit TIRF imaging data at t = 1 min. (**C and D**) Decay kinetics of the fraction of the ULBP3-NKG2D complex when NKG2D molecules are distributed in space uniformly (magenta asterisks, Model 1; magenta diamonds, Model 2) or in clusters obtained from NKG2D configuration at t = 1min for (C) Model 1 and (D) Model 2. NKG2D molecules are drawn from a uniform random distribution for creating the homogeneous configurations where a 0.90 (0.83) fraction of NKG2D are bound to ULBP3 at t = 0 for Model 1 (Model 2). The decay kinetics are averaged over 200 initial configurations. The solid lines show fits to exponential decays at early times. The decay kinetics start deviating from the exponential decay at t ≳ 10sec.

movements by an actin polymerization inhibiting drug cytochalasin D produces a substantial decrease in pVav1 in NKLs stimulated by NKG2D ligand MICA. Actin polymerization induces microcluster formation and centripetal movements of NKG2D as treatment with actin depolymerization agent latrunculin abrogated NKG2D microcluster formation and movements in NKLs [6]. Next, we used Model 2 for deciphering mechanisms of signal integration.

### 3. pVav1 dependent centripetal movements of NKG2D are abrogated by inhibitory KIR2DL2 signaling

We investigated inhibition of NKG2D signaling by inhibitory KIR2DL2 receptors in Model 2. The KIR2DL2 receptors were distributed in the simulation box following TIRF imaging data in ref. [6] (S5A Fig). The two-point correlation function calculated from spatial distribution of KIR2DL2 at t = 0 in our simulations agrees well with that of the TIRF imaging data (S5C Fig). The TIRF experiments show that KIR2DL2 microclusters are present in higher numbers at the periphery compared to the central region of the IS. In addition, the spatial organization of these microclusters does not change appreciably between pre- and ~30s post- stimulation by HLA-C ligands (S5B Fig). The changes in KIR2DL2 microcluster distributions beyond 30s occurred presumably due to rapid retraction of NK

cells from the supported lipid bilayer. Since we did not model NK cell retraction in our models, we assumed KIR2DL2 microclusters to be stationary in our simulations. The chambers where the number of KIR2DL2 exceeded a threshold value were considered to be parts of KIR2DL2 microclusters in simulations. In our models, KIR2DL2 microclusters co-localize with SFK which is supported by previous experiments [54]. NKG2D, ULPB3, HLA-C, SFK, SHP1, and Vav1 were distributed homogeneously in the simulation box at t = 0. Some of the SFK molecules were co-clustered with KIR2DL2 which remained immobile for the duration of the simulations (details in Materials and Methods section). In the simulations, SHP1 recruited by the pITIMs in KIR2DL2-HLA-C complexes dephosphorylate pVav1 (bound to NKG2D complex or free) residing in the same spatial location or the same chamber. Thus, the production of pVav1 reduces substantially in the presence of inhibitory KIR2DL2 signaling (Fig 5B). The decrease in pVav1 abundance also slows down centripetal movements of the NKG2D microclusters resulting in a more spread-out spatial distribution of NKG2D in simulations (Fig 5A and S3 Movie). We quantified the reduction of centripetal migration of NKG2D by calculating the increase in the number of NKG2D molecules in a region enclosing the center of the IS (Fig 5C) over a time period. This decrease in the centripetal movement of NKG2D is qualitatively in agreement with experiments by Abeyweera et al. [6].

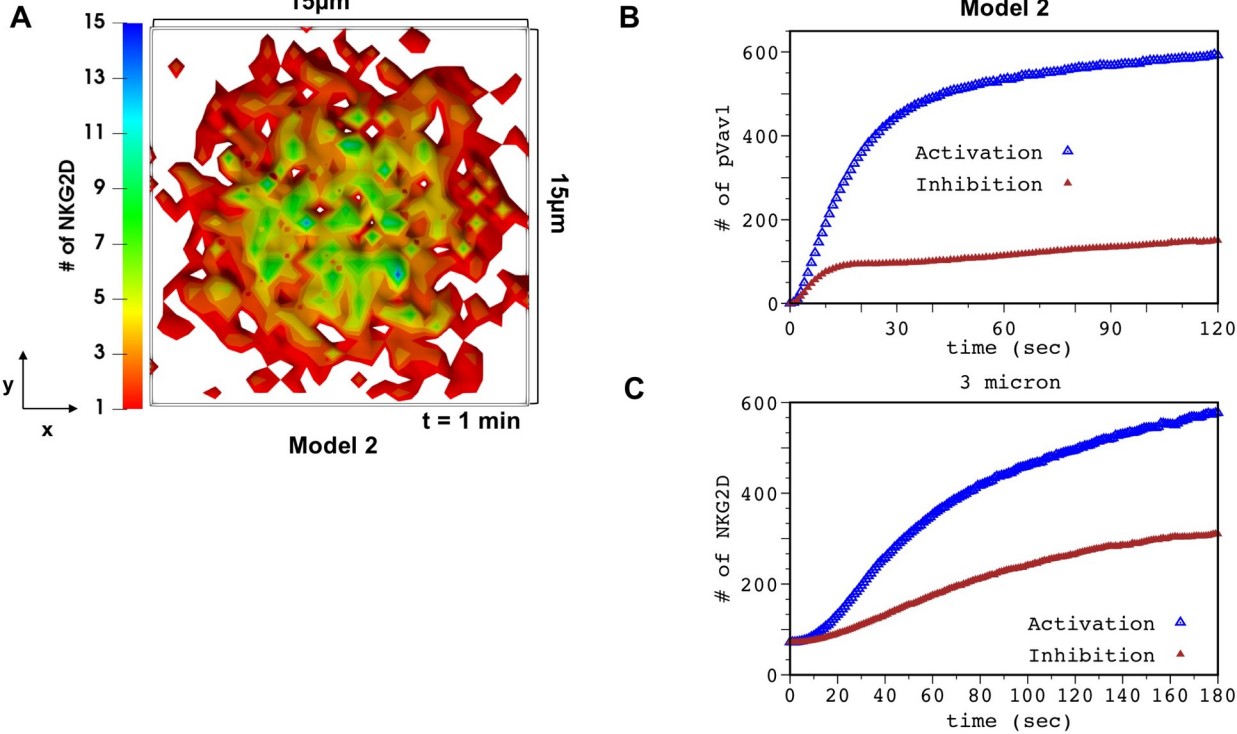

**Fig 5. KIR2DL2 inhibition abrogates centripetal movements of NKG2D receptor clusters. (A)** Spatial clustering of NKG2D in the presence of inhibitory KIR2DL2 signaling in Model 2 at t = 1 min post ULBP3 stimulation. KIR2DL2 is stimulated by HLA-C at t = 0. KIR2DL2 is distributed in the simulation box following the data extracted from TIRF imaging at 1 min 44 sec in Fig 4 of Ref. [6]. **(B)** Kinetics of total number of pVav1 in Model 2 in the presence (filled brown triangle) and the absence (empty blue triangle) of inhibitory ligands (HLA-C). **(C)** The kinetics of the number of NKG2D molecules in a region of area 3μm ×3μm around the center of the simulation box for Model 2 in the presence (filled brown triangle) and absence (empty blue triangle) of inhibitory ligands (HLA-C). The decrease in the number of NKG2D in the presence of KIR2DL2 signaling shows the decrease in the centripetal flow of the NKG2D microclusters in the simulation. The pVav1 and NKG2D kinetics shown are averaged over 200 different configurations.

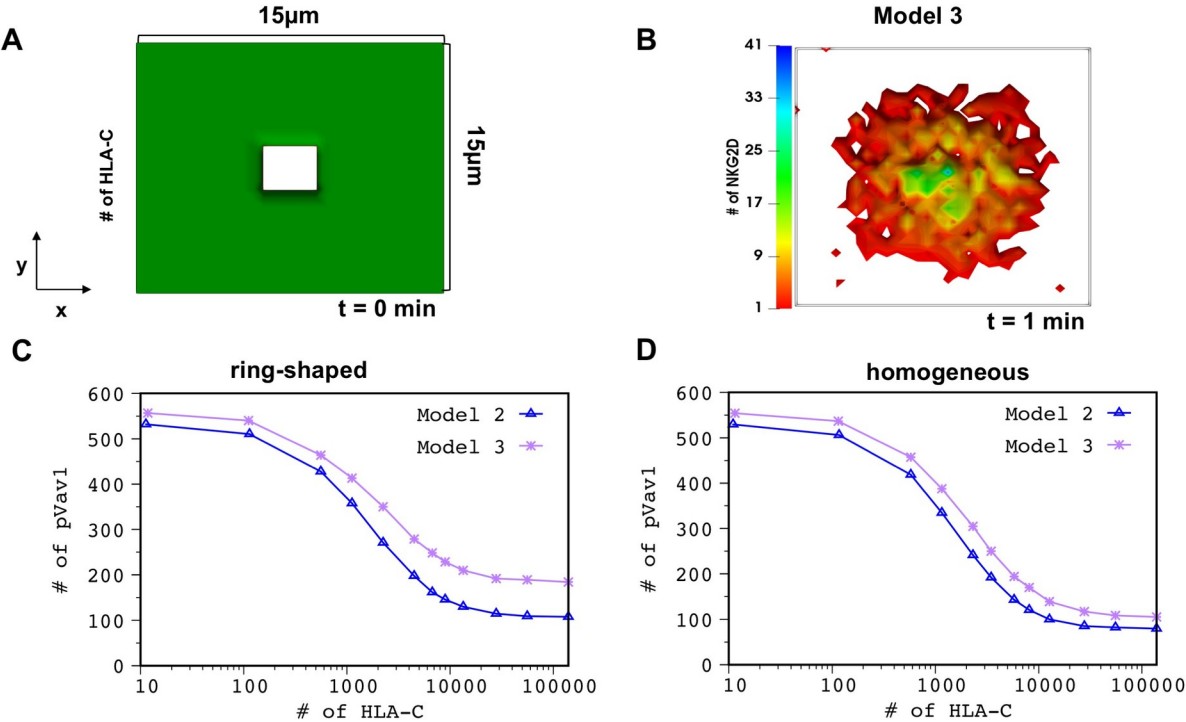

**Fig 6. The presence of the spatial positive feedback allows for efficient inhibition. (A)** Shows spatial arrangement of inhibitory HLA-C ligands in a ring configuration used in the simulations of Model 2 and Model 3. **(B)** Spatial distribution of NKG2D the simulation box for Model 3 at 1 min in the presence of both activating ULBP3 and inhibitory ligands shown in (A). Note the central accumulation of NKG2D because of the independence of NKG2D centripetal movements on pVav1 for Model 3. **(C and D)** Show variation of total number of pVav1 at t = 1 min with increasing HLA-C concentrations for Model 2 (empty blue triangle) and Model 3 (purple asterisks). The decrease in pVav1 is higher in Model 2 compared to Model 3. The decrease in pVav1 Model 2 is more pronounced at larger HLA-C concentrations when HLA-C are organized in a ring pattern (C) vs distributed homogeneously (D). The pVav1 values shown in (C) and (D) were obtained by averaging over 50 configurations for each HLA-C dose.

## 4. Interplay between Vav1 phosophorylation and centripetal movements is required for efficient inhibition of NKG2D signaling by inhibitory KIR2DL2

We employed Model 2 and Model 3 to study the role of the interplay between centripetal movements of NKG2D microclusters and pVav1 in early time integration of NKG2D and KIR2DL2 signals. In Model 3, centripetal movements of NKG2D and Vav1 are independent of pVav1 abundances, thus, the decrease in pVav1 abundances due to inhibitory signaling does not affect accumulation of NKG2D at the central region of the IS (S4 Movie and Fig 6B). We carried out simulations for two scenarios where inhibitory HLA-C ligands were distributed in the simulation box homogeneously (Case I) or in a ring pattern (Case II) (Fig 6A) devoid of HLA-C molecules at the center of the ring. The ring pattern could represent HLA molecules on engineered surfaces [55] or on target cells. Almeida et al. [54] found HLA-C molecules to be organized in a ring pattern on target cells after NK cells are incubated with target cells for 10mins. The pattern formation in part is initiated by binding of inhibitory KIR2DL2 and HLA-C, and KIR2DL2 co-localizes with HLA-C in such clusters. Simulations were carried out with HLA-C and ULBP3 in Model 2 and Model 3. When HLA-C are distributed homogeneously (Case I), the total number of pVav1 at t = 1 min post NKG2D stimulation decreases substantially in both models as the number of HLA-C increased about 4 -fold (1000 to 4000 in the simulation box) (Fig 6D). The pVav1 abundances are slightly higher in Model 3 than

Model 2 owing to increased NKG2D clustering in the former model. This decrease in pVav1 with increasing HLA-C in the models is qualitatively similar to the large decrease in the percentage of lysis of human NK cell line YTS in cytotoxicity assays reported in ref. [54] as the abundance of HLA-C on target cells increased about 4 -fold. Next, we investigated variation of pVav1 with increasing HLA-C dose in Model 2 and Model 3 when HLA-C molecules were distributed in a ring pattern (Case II). The simulations show smaller decrease in the number of pVav1 in Model 3 compared to Model 2 as the number of HLA-C is increased (Fig 6C). This result can be explained as follows. In Model 3, NKG2D microclusters, regardless of the pVav1 concentrations, accumulate at the center of the IS devoid of HLA-C and produce pVav1 even at high HLA-C concentrations, whereas, in Model 2 the lower amount of pVav1 at high HLA-C concentrations decreases centripetal movements of NKG2D and prevents accumulation of NKG2D at the center of the IS (S6 Fig). We also carried out the above comparison when KIR2DL2 molecules are co-localized with HLA-C in the ring pattern to represent HLA-C clustering in target cells interacting with NK cells [54]. The results (S7 Fig) were similar to that in Fig 6C. Therefore, the HLA-C ligands are able to suppress the production of pVav1 more efficiently in Model 2 compared to Model 3.

## Discussion

Spatial organization of activating human NKRs such as CD16[56], NKp46[57], KIR2DS1[58], and NKG2D [5–7], and associated signaling proteins has been observed in NK cells stimulated by cognate activating ligands. Microcluster formation by NKRs can increase the apparent lifetime of NKR-ligand complexes because of higher frequency of ligand rebinding within a microcluster. In addition, co-clustering of NKRs with signaling proteins in NKR microclusters can increase biochemical propensities of signaling reactions due to elevated local concentrations of reacting proteins. Either of the above mechanisms could increase production of activated (e.g., tyrosine phosphorylated) signaling proteins. However, when the contribution due to ligand rebinding is not substantial, co-clustering of NKR and other signaling proteins plays a dominant role in increasing downstream signaling. We developed a predictive in silico framework to quantify relative roles of the above mechanisms for early (~ few mins) NKG2D signaling kinetics to demonstrate that co-clustering of Vav1 with NKG2D plays a more dominant role over ULBP3 rebinding in increasing activating signals. Since NKG2D binds with ULPB3 with a half-life ($\ln(2)/k_{off}$) of ~ 30 seconds and the average number of homogeneously distributed NKG2D molecules in the area occupied by a typical NKG2D microcluster is about two molecules, the higher numbers of NKG2D in microclusters do not result in a large increase in abundances of NKG2D-ULBP3 complexes due to ligand rebinding. Our simulations confirm the above explanation. Co-clustering of signaling proteins could play an important role for increasing downstream signaling for other activating NKRs as well. The density of CD16 on primary NK cells is about 200 times larger than that of NKG2D [59] and the lifetime of a CD16-ligand complex is similar (e.g., $k_{off} < 0.01$ s$^{-1}$ for IgG[60]) or larger than that of NKG2D-ULBP3, therefore, the effect of co-clustering of signaling proteins with CD16 could be more relevant for increasing downstream signaling than ligand rebinding.

Our NKG2D signaling model could describe spatiotemporal clustering by other activating NKRs such as CD16 qualitatively, as the formation and movements of another activating NKR microclusters could arise from similar interplay between pVav1 kinetics and cytoskeletal reorganizations modeled here, however, the time scales of formation and centripetal movements of microclusters of activating NKRs would depend on the time scales of pVav1 production. Since, the signaling reactions leading to pVav1 production can be different between NKG2D

and another activating NKR (e.g., CD16, KIR2DS1), the spatiotemporal kinetics of NKG2D and other activating NKRs are likely be quantitatively different.

Our agent based model (Model 2) is successful in generating predictions regarding inhibition of NKG2D signaling by inhibitory KIR2DL2 signaling or by drugs inhibiting actin polymerization. The model qualitatively captures the slowing down of the NKG2D microcluster movements in the presence of inhibitory KIR2DL2 signaling (Fig 5A and 5C) and the decrease in pVav1 (Fig 5B) in response to an actin polymerization inhibiting drug cytochalasin D in NKLs stimulated by NKG2D ligand MICA [14]. The model also quantitatively predicts spatial pattern of NKG2D in TIRF imaging at later time points, not included in model training, reasonably well. The model also reasonably predicted (S8 Fig) early ($\leq$ 70 secs) clustering of NKG2D in NKLs stimulated by MICA on target Daudi cells in confocal experiments reported by Brown et al. [7]. This provides confidence on the applicability of our model and parameter estimation for describing other systems. The model can be further expanded to investigate mechanisms of interplay between pairs of activating and inhibitory NKRs stimulated by activating ligands and different organizations of HLA-C ligands with peptides. Adaptable spatial patterns of antigens presented by DNA origami structures have been recently used to manipulate responses in B- cells [61] and CAR T- cells [62]. The spatial model developed here can serve as a screening tool for identifying spatial patterns of antigens optimal for generating specific lymphocyte responses.

In order to investigate the utility of our model to predict further downstream NK cell responses, we applied our model to predict lysis of target cells treated with DNA polymerase inhibitor aphidicolin in NK cell cytotoxic assays by Gasser et al. [63]. Gasser et al. [63] observed aphidicolin treatment moderately increases the expression of NKG2D ligands in mouse T-cell blast target cells and these target cells showed increased lysis compared to their untreated counterparts. An extension of our signaling model qualitatively predicts (S3 Text and S16 Fig) the differences in lysis of aphidicolin treated and untreated target cells for a range of effector cell:target cell ratios as reported by Gasser et al. [63]. This application could point to a potential way to combine early time signaling model with population kinetic models describing later time NK cell responses.

We studied the role of the interplay between creation of NKG2D microclusters, centripetal movement of the microclusters, and early time signaling using an alternate model (Model 3) where centripetal movement of NKG2D microclusters is independent of pVav1. Our simulations show that pVav1 dependent centripetal movements allow for a more efficient suppression of pVav1 in the presence of inhibitory KIR signaling, in particular, when inhibitory ligands (HLA-C) are clustered in a ring formation in the target cell membrane. HLA-C molecules have been found to organize in ring or multifocal patterns on target cells [54], the presence of the spatial feedback in NKG2D microclusters and pVav1 creates an efficient and flexible maneuver to inhibit signaling when inhibitory ligands are present in spatially inhomogeneous patterns.

A unique aspect of our modeling framework is estimation of model parameters that best fit spatial patterns of NKG2D measured in TIRF imaging experiments. Many parameters in our model were not measured in experiments previously or are difficult to measure because they described coarse-grained processes (e.g., parameters in describing centripetal force). In addition, reported parameter values measured under specific conditions (e.g., outside the cell) might not reflect its magnitude within the cell. Therefore, we carried out an estimation of model parameters using parallel codes that produced quantitative agreement between spatial patterns of NKG2D in simulation and imaging experiments. We used two-point correlation functions to quantify spatial organization of NKG2D microclusters which is widely used in statistical physics and materials science to quantify spatial patterns. The parameter estimation

step provides the following benefits: (i) it allows models to capture unique aspects of kinetics of NKG2D microclusters, e.g., centripetal movement with decreasing mobility at the center of the IS. (ii) Provides parameter ranges where the roles of ligand rebinding and co-clustering of NKG2D and Vav1 can be compared. Since both the processes depend on parameter values, these should be compared in parameter ranges that are able to reproduce experimental data. Receptor clustering in the IS has been widely investigated in computational models in T-[64–70], B-[71], and NK- cells[11,12], however, most of these models described the spatial patterns qualitatively. Some of the modeling quantitatively described receptor patterns in NK-[72] and T-[73] cells, however, did not couple receptor clustering with signaling kinetics. A major difficulty in parameter estimation in spatial models is computationally intensive nature of such calculations, which we address here using parallel computation. However, as we discuss below there are several areas where this framework requires further improvement.

## Limitations of the work

The model did not include several processes or made approximations to keep the model simple and focused on the questions of interest addressed here. We did not include integrin receptor LFA1 and its ligand ICAM-1 in our models for simplicity and focused our study to NKG2D and KIR2DL1 signaling and clustering. Larger size LFA1-ICAM-1 complexes (~40nm) segregate from the smaller sized (~10-15nm) NKG2D-MICA or KIR2DL1 molecules due to size based exclusion interactions [74]. Usually, LFA1-ICAM1 molecules reside in the periphery of the IS surrounding NKG2D [74,75]. However, integrin receptor signaling contributes towards NK cell signaling [76–78] and spatial clustering of LFA-ICAM1 will likely contribute towards early time NK signaling and its regulation of NKG2D spatiotemporal signaling kinetics. Similarly, proximal signaling proteins such as PI3K [79] were not included in the model to keep the model simple and focused on the questions of interest addressed here. The models also do not include NKG2D degradation [80] which tends to occur later in the signaling (>15 mins)[81] and could underlie the lower amount of phosphorylated ITAM phosphorylation in the central region in the TIRF experiments in Abeyweera et al. [6]. We represented multiple SFKs (e.g., Lck, Fyn, Lyn, Yes) in NK cells by a single species for simplifying models. There are differences in substrate specificities [82] for SFKs in T cells. It will be straightforward to extend our models to include multiple SFKs if similar differences are found in NK cells.

Furthermore, our model did not include spreading of the NK cell surface on the glass slide. Upon NKG2D stimulation the NK cell spreads at later times ($\geq$ 4min). Due to this reason our model can describe early time (0–3 min) NKG2D spatiotemporal signaling kinetics where NK cell spreading is negligible but is unable to quantitatively predict changes in NKG2D spatial patterns due to spreading of NK cell surface on the glass slide at times $\geq$ 4min (S3 Fig) or late time signaling events such as secretion of lytic granules. The model also does not include retraction of the NK cell surface from the bilayer within minutes (~ 5 mins) induced by stimulation of inhibitory KIR2DL2 in the TIRF experiments [6]. The retraction helps to self-limit the formation of NK cell-target cell conjugates when inhibitory signals dominate [5,6].

One the computation side, several parameters showed large error bounds implying these parameters can be changed substantially but will make small or no changes to the mean cluster sizes and two-point correlation functions. A potential solution to address this challenge could be introducing weight factors for combining mean values and two-point correlation functions (or second moments) following a systematic framework such as generalized method of moments [83,84].

## Materials and methods

### Spatial kinetic Monte Carlo simulation

We used the software package SPPARKS (https://spparks.sandia.gov/) to simulate reactions and particle hopping moves described below. We chose Gibson-Bruck implementation of Gillespie algorithm [85] within SPPARKS to perform the kinetic Monte Carlo simulation.

### Biochemical reactions

Molecules in each chamber of volume ($l_0 \times l_0 \times z$) where $l_0 = 0.5\mu m$ and $z = l_0$ (for plasma membrane bound molecules) or $z = 2l_0$ (for cytosolic molecules) are taken to be well-mixed. Stochastic biochemical reactions in individual chambers are simulated using reaction propensities listed in Table 1. There are three types of protein molecules depending on their location in the simulation box: (i) Molecules residing in the NK cell plasma membrane which include NKG2D, SFK, and, KIR2DL2. (ii) Molecules residing in the NK cell cytosol which include Vav1, and SHP1. (iii) Molecules residing in the supported lipid bilayer in TIRF experiments which include NKR ligands: ULBP3, and HLA-C. The binding of NKRs and their respective ligands can occur when the binding domains of these proteins are separated by short distances $d_{recep-lig} \sim 2$ nm [86]. Therefore, the volume factor ($v_{recep-lig}$) used for converting the unit of binding rate ($k_{on}$) from $(\mu M)^{-1} s^{-1}$ to propensity ($\propto k_{on}/v_{recep-lig}$) in $s^{-1}$ in a chamber is taken as, $v_{recep-lig} = l_0 \times l_0 \times d_{recep-lig}$. Similarly, the volume factor $v_{recep-SFK}$ used for binding of NK cell receptors with SFKs is set to, $v_{recep-SFK} = l_0 \times l_0 \times d_{recep-SFK}$, where $d_{recep-SFK} \sim 10$ nm [16]. The volume factor $v_{cytosol}$ for cytosolic molecules binding with NK cell plasma membrane bound complexes is taken as, $v_{cytosol} = (l_0 \times l_0 \times 2l_0)$. In Model 2, the volume factor for binding of plasma membrane associated Vav1 molecules with other plasma membrane residing molecules is taken as, $v_{Vav1-plasma} = (l_0 \times l_0 \times l_0)$.

### Microcluster formation

A pVav1 dependent processes is implemented to generate microclusters of NKG2D. A "potential" function $E_i$ is associated with any chamber $i$, is given by,

$$E_i = \min\left(-x_i^2, \quad -x_{j1}^2, \quad -x_{j2}^2, -x_{j3}^2, -x_{j4}^2\right)$$

where, $x_i$ is the number of pVav1 molecules in the $i^{th}$ chamber and $\{j1, j2, j3, j4\}$ denote its four nearest neighbors.

The probability of moving NKG2D complexes from the $i^{th}$ chamber to its nearest neighbor $j$ is given by, $p^{(m-clus)}_{ji} = min\left\{1, \frac{e^{\beta\left(E_i - E_j\right)}}{1 + e^{\beta\left(E_i - E_j\right)}}\right\}$; thus, when a nearest neighbor $j$ has a potential lower than the $i^{th}$ chamber, i.e., $E_j < E_i$, NKG2D molecules move to the $j^{th}$ chamber. We estimate $\beta$ using PSO.

### Diffusive movements

Unbound molecules of NKRs, their cognate ligands, SFK, SHP1, and Vav1 move diffusively. The diffusive movements are simulated by hopping moves of molecules to the nearest neighbor chambers with probability $p^{diffu} \propto D/(l_0)^2$. The propensities for these movements are given in Table 1. Periodic boundary conditions in the x-y plane are chosen for freely diffusing molecules. The propensity for diffusive movements of cytosolic molecules is about >1000 fold larger than that of the plasma membrane bound molecules and that of most reactions. This is because of the larger value of the diffusion constant (~10 $\mu m^2/s$) [87] and the larger average

number of cytosolic molecules (~ 30–200 molecules) in a chamber. Therefore, a substantial proportion of the Monte Carlo moves are spent on diffusive moves of cytosolic molecules which make the run time of the simulation long (wall time ~10 hrs on a single 3.0 GHz AMD EPYC processor). We approximated these fast diffusive moves by not executing explicit diffusion moves for the cytosolic molecules but by homogenizing cytosolic molecules in the simulation box at time intervals proportional to the average time a cytosolic molecule will need to diffuse the length of the simulation box. We checked the validity of this approximation by comparing simulations incorporating the above approximation with those containing explicit diffusive movements for cytosolic molecules. We did not find appreciable differences between the two (S9 Fig).

## Microcluster movements

Any ULBP3 bound NKG2D receptor complex is assumed to be a part of a microcluster. All molecules of a specific ULBP3-NKG2D complex, e.g., ULBP3-NKG2D-SFK, in a chamber are moved to one of the four nearest neighboring chambers in a single microcluster hopping move. The probabilities of the hops to the neighboring chambers are given by $\vec{p}_{\text{hop}} \equiv (p_{\text{right}}, p_{\text{left}}, p_{\text{up}}, p_{\text{down}})$ and the corresponding propensities are calculated by multiplying the probabilities by a rate $k^{(\text{cluster-move})}$. $\vec{p}_{\text{hop}}$ at a position (x,y) in the x-y plane is given by,

$$p_{left/right} = \frac{1}{4}\left(1 \pm s\left(\frac{x}{R} - 1\right)\right)w + \frac{1}{4}(1 - w)$$

$$p_{down/up} = \frac{1}{4}\left(1 \pm s\left(\frac{y}{R} - 1\right)\right)w + \frac{1}{4}(1 - w) \tag{1}$$

, where, the four corners and the center of the simulation box in the x-y plane are at (0, 0), (0, 2R), (2R,0), (2R, 2R), and (R,R), respectively. The variable $s$ depends on the total number of pVav1 (or $[pVav1]_T$) in the simulation box and is given by,

$$s = \frac{[pVav1]_T}{K + [pVav1]_T}$$

The hopping probability $\rightarrow p_{\text{hop}}$ is composed of two parts: one generates centripetal movements[88, 89] and the other produces random Brownian movements. The weight factor $0 \leq w \leq 1$ determines the relative proportions of centripetal and random components in $\vec{p}_{\text{hop}}$, i.e., the smaller the $w$, the stronger is the bias toward random movements. The velocities for centripetal movements along x and y directions are given by,

$$v_x = sw\left(\frac{x}{R} - 1\right), \ v_y = sw\left(\frac{y}{R} - 1\right).$$

Therefore, the magnitude of the centripetal velocity $v_r$ is given by,

$$v_r = sw\sqrt{\left(\frac{x}{R} - 1\right)^2 + \left(\frac{y}{R} - 1\right)^2}.$$

$v_r$ decreases monotonically from the periphery to the center of the simulation box. $v_r = 0$ at the center (R,R) and is maximum ($\sqrt{2}sw$) at the corners of the simulation box. In the absence of any pVav1 in the simulation box, the centripetal movements cease to exist, i.e., $v_r = 0$. Velocity of F-actin retrograde flow decreases monotonically from the periphery to the center in YTS NK cell lines stimulated by surface coated anti-NKG2D[15]. We assume that NKG2D

microcluster movements are guided by F-actin, similar to that to TCR microclusters in antigen stimulated Jurkat T cells [90].

The rate, $k^{(cluster-move)}$ determines the magnitude of the centripetal velocity and the diffusion constant of the Brownian movements of microclusters. A movie of movements of a single microcluster is shown in the supplementary material (S5 Movie).

### Excluded volume interactions

We impose an upper bound ($N_{thres}$) to the number of molecules that can reside in a chamber to incorporate excluded volume interactions. Molecules are allowed to hop to a chamber if the number of molecules does not increase beyond $N_{thres}$ due to the move. A separate upper bound for the number of NKG2D in a chamber is implemented to prevent aggregation of NKG2D to a small number of very high density NKG2D microclusters in the central region.

### Influx of NKG2D

An influx of NKG2D molecules in the simulation box from the boundary with a rate $k^{(influx)}$ is implemented by adding an outer rim of thickness $2l_0$ at the boundary of the simulation box. The chambers in the rim are occupied by NKG2D and ULBP3 which undergo NKG2D-ULPB3 binding-unbinding reactions. The NKG2D-ULBP3 complexes do not participate in further downstream reactions inside the outer rim. The bound NKG2D-ULPB3 complexes enter the simulation box by hopping moves that occur at rate $k^{(influx)}$.

### Initial configuration

Unbound molecules of NKG2D, SFK, Vav1, KIR2DL2, and SHP1 were distributed homogeneously in the simulation box where any chamber contained the same number of molecules for a particular protein. The numbers of molecules in each chamber for the above protein species are calculated using the values shown in Table 1 assuming proteins are distributed homogeneously in the plasma membrane or the cytosol of an NK cell. The number of ULBP3 in the simulation box is estimated via PSO, where, a chamber at $t$ = 0 is populated with 3 molecules (or left unpopulated) of ULBP3 with a probability $f$ (or 1-$f$). $f$ is estimated in PSO.

### Particle swarm optimization

We performed asynchronous particle swarm optimization (PSO) algorithm with constraints to optimize a cost function to an estimate model parameters. The parameters were varied as powers of 10 in PSO. The parameters used in the PSO are the following: particle velocity scaling factor, $\omega$ = 0.5, coefficient weighting a particles best-known position, $\varphi_p$ = 2.5, and coefficient weighting the swarm's best-known position, $\varphi_g$ = 1.5. A swarm size of 200 particles is considered. The maximum iteration limit is assigned to 100. The computation takes about 48h on 300 parallel 3.0 GHz AMD EPYC CPUs. A similar set of PSO parameters were used to estimate parameters in a spatial model describing formation of bacterial biofilms in three dimensions [91]. The construction of the cost function is described below.

### Construction of the cost function

We extracted intensity profile I({$x_i, y_i$},t) of NKG2D-DAP10 using TIRF images (details in *Extraction of image data*), where {$x_i, y_i$} denote the centers of the chambers in the x-y plane in the simulation box. Our parameter estimation scheme minimizes a cost function that measures the Euclidean distance between variables quantifying statistical properties of spatial patterns of NKG2D in TIRF experiments and simulations from our agent based models. We computed

mean,

$$\mu_S(t) = 1/N_{chamber} \sum_i S(x_i, y_i, t), \tag{2a}$$

variance,

$$\sigma_S^2(t) = 1/N_{chamber} \sum_i (S(x_i, y_i, t) - \mu_S(t))^2, \tag{2b}$$

and two-point correlation function [52,91],

$$C_s(r,t) = 1/N_{chamber} \sum_i \sum_{r_x, r_y: r_x^2 + r_y^2 = r^2} (s(x_i, y_i, t) - \mu_s(t))(s(x_i + r_x, y_i + r_y, t) - \mu_s(t)) \tag{2c}$$

The summation over $r_x$ and $r_y$ indicates average of all neighbors of $(x_i, y_i)$ separated by a distance $r$, i.e., $r_x^2 + r_y^2 = r^2$. We used periodic boundary conditions for the calculation of $C_S(r,t)$ for $r \le L/2$, where $L$ is the length of the simulation box. For the TIRF imaging data, $S(\{x_i, y_i\}, t) = I(x_i, y_i, t)/I_{max}(t)$, and for our model simulations, $S(\{x_i, y_i\}, t) = n(\{x_i, y_i\}, t; \theta)/n_{max}(t)$, where $n(x_i, y_i; t; \theta)$ denotes the number of NKG2D molecules in a chamber centered at $(x_i, y_i)$ in the x-y plane. $\theta$ represents model parameters listed in Table 1. $I_{max}(t)$ and $n_{max}(t)$ denote the maximum values of image intensity and NKG2D number in the chambers in the imaging data and in the simulation, respectively.

The quantities in Eq (2) depend on the initial state of the system at $t = 0$ in simulations or experiments and on intrinsic noise fluctuations that arise during time evolution of the system (S10 Fig). Averages of these quantities (denoted by $\langle \cdots \rangle$) over ensembles of configurations at time $t$ arising from different initial states and stochastic trajectories with intrinsic noise fluctuations are traditionally used in physics[52] to characterize spatial patterns. However, it is challenging to perform such ensemble averages in our situation because, (i) few replicates of experimental data are usually available, and (ii) computational cost of performing ensemble averaging within PSO. In order to circumvent these issues, we minimized a cost function as a function of the quantities in Eq (2) where the best-fit values of the model parameters depend on the initial state as well as on the intrinsic noise fluctuations. However, we account for effect of the intrinsic noise fluctuations in the parameter estimation as described in the error estimation section.

The cost function is given by,

$$C_{cost}(\theta; \eta; n_0) = \frac{1}{3}\left(\frac{\mu_I - \mu_n}{\mu_I}\right)^2 + \frac{1}{3}\left(\frac{\sigma_I - \sigma_n}{\sigma_I}\right)^2 + \frac{1}{3}\sum_r \left(\frac{C_I(r,t)}{C_I(0,t)} - \frac{C_n(r,t)}{C_n(0,t)}\right)^2 \tag{3}$$

$\eta$ represents random variables associated with the intrinsic noise fluctuations and $n_0$ denotes the initial state at $t = 0$ in the model. The minimization of Eq (3) in our PSO yields a $\theta = \theta_{min}$ associated with a specific set of random variables $\eta_{pso\text{-}optim}$ and a fixed initial state $n_0$, i.e., the minimum value of $C_{cost}$, $C_{cost}|_{min} = C_{cost}(\theta_{min}; \eta = \eta_{pso\_otim}; n_0)$.

## Quantification of uncertainties in PSO estimation of parameters

We generated configurations $\{n(\{x_i, y_i\}, t; \theta = \theta_{min})\}$ for the best-fit parameter value $\theta_{min}$ for an initial state $n_0$, and different sets of intrinsic noise fluctuations (or different $\eta$) in our simulations. We computed $C_{cost}(\theta_{min}; \eta, n_0)$ given by Eq (3) for each $n(\{x_i, y_i\}, t; \theta_{min})$ and computed the variance, $\sigma_C$, for the values of $C_{cost}$ evaluated for the ensemble, $\{n(\{x_i, y_i\}, t; \theta)\}$ (S11 Fig). We reasoned that the range $0 \le C_{cost} \le C_{cost}|_{min} + 2\sigma_C$, will be scanned by intrinsic noise fluctuations in the model for $\theta_{min}$; therefore, a configuration $n(\{x_i, y_i\}, t; \theta)$ where $\theta \neq \theta_{min}$ that generates a cost function $C_{cost}$ within above range cannot be separated well from $\{n(\{x_i, y_i\}, t; \theta_{min})\}$. This

produces an uncertainty in our estimated parameters $\theta_{min}$. We characterized the uncertainty in $\theta_{min}$ by collecting the parameters $\{\theta\}$ that produce cost functions in range 0 to $C_{cost}|_{min} + 2\sigma_C$ (S11 Fig). Next, we evaluated if these parameters reside in a single or multiple clusters in the manifold spanned by the parameters–the existence of multiple clusters will indicate the presence of multiple local minima. We computed the number of cluster following a method in Ref. [92] and found a single cluster (S12 Fig). Each parameter was scaled as $(\theta - \theta_{min})/(\theta_{max} - \theta_{min})$, where $\theta_{min}$ and $\theta_{max}$ are the minimum and the maximum parameter values, respectively, to make it dimensionless and to lie between 0 and 1 before applying the algorithm in Ref. [92]. The standard deviations computed for the points in the cluster gives an estimate of the uncertainty in $\theta_{min}$.

### Extraction of image data

Intensities for fluorescently labeled molecules in TIRF experiments were extracted from images published in in Ref. 6. The python code used for the extraction is uploaded on github at https://github.com/jayajitdas/NK_signaling_spatial_model. The extracted intensities for fluorescently tagged NKG2D-DAP10 or KIR2DL2 molecules from regions of interest were averaged over multiple pixels to create spatial data with the minimum resolution (~0.5 μm) in our simulation. Further details are provided in S13 Fig.

### Supporting information

**S1 Fig. Comparison of $\mu_I$ and $\sigma_I$ for TIRF images with $\mu_n$ and $\sigma_n$ for configurations in model simulations.** Shows $\mu_I$ and $\sigma_I$ calculated from TIRF images (S4 Fig in Ref. [6]) and model simulations at t = 1, 2, 3, 4, 5, 6, and 7 mins. The calculations of $\mu_I$ and $\sigma_I$ are shown in Eq 2a and 2b in the main text. The values for TIRF images and model simulations are shown along the y and x axes, respectively. The x = y line is shown to quantify agreement/deviation between imaging data and models. The symbols indicated by $\mu_a$ and $\sigma_a$ depict the values of ($\mu_I$, $\mu_n$) and ($\sigma_I$, $\sigma_n$) in the x-y plane at times a = t1 to t7 denoting times 1 to 7 mins, respectively. Comparisons are shown for (**A**) Model 1 and (**B**) Model 2. The models are simulated for the best-fit PSO parameters.
(TIF)

**S2 Fig. Ensemble averaged two-point correlation function for Model 1 and Model 2 at multiple time points.** Shows comparisons between ensemble averaged two-point correlation function ($<C(r,t)/C(0,t)>$ vs r) for Model 1 (black, empty circle) and Model 2 (blue, empty triangle) with C(r,t)/C(0,t) calculated from TIRF image (red, empty square) at (**A**) t = 1min, (**B**) t = 2 min, and (C) t = 3 min. The parameters for the simulation are set at the best-fit values from the PSO. The two-point correlation functions for the models are averaged over an ensemble of 200 configurations.
(TIF)

**S3 Fig. Model predictions for NKG2D spatial pattern at t = 4 min show deviations from the TIRF imaging data.** Shows comparison between the two-point correlation function (C(r, t)/C(0,t) vs r) at t = 4 min calculated from TIRF image (red, empty square) and configurations simulated by Model 1 (black, empty circle) and Model 2 (blue, empty triangle). The parameters for the simulations are set to the best-fit values obtained from our PSO. The models show deviations from the TIRF imaging data for length scales 1–3 μm which can potentially arise due to substantial spreading of the NK cell on the lipid bilayer at 4 mins.
(TIF)

**S4 Fig. Effect of formation of microclusters on pVav1 production kinetics for Model 1 and Model 2.** Shows increase in the total number of pVav1 with time in Model 1 (**A**) and Model 2 (**B**) when NKG2D are not allowed to form microclusters (magenta asterisks, Model 1; magenta diamonds, Model 2) or form microclusters (black open circles, Model 1; blue open triangles, Model 2) according to the model rules. The pVav1 concentrations are averaged over 200 different configurations. The models are simulated for the best-fit PSO parameters.
(TIF)

**S5 Fig. Analysis of spatial distribution of KIR2DL2 receptors extracted from TIRF imaging data. (A)** Shows initial configuration of KIR2DL2 at t = 0 in our simulations based on the coarse-grained 2D image extracted for a region of interest in TIRF image (Fig 4 in Ref. [6]) of KIR2DL2-GFP at t = 1 min 44 sec to match the minimum length scale (~ 0.5 μm) of spatial resolution in our model. (**B**) C(r,t)/C(0,t) calculated for intensities of KIR2DL2-GFP extracted from TIRF experiments (Fig 4 in Ref. [1]) at times prior and ~30s post stimulation by HLA-C ligands (UV irradiation at t = 1min 44 seconds (00:01:44), blue filled circles) for the region of interest from TIRF images. (**C**) C(r,t)/C(0,t) calculated for intensities of KIR2DL2-GFP for the region of interest from TIRF image (blue empty triangles) and after coarse-graining (corresponding to image A) to obtain the minimum scale length resolution of our simulation box (green cross).
(TIF)

**S6 Fig. Accumulation of NKG2D in the central region of the IS in the presence of pVav1 dependent/independent centripetal motility.** Shows the number of NKG2D molecules in a 3μm ×3μm area at the center of the simulation box for Model 2 (M2) and Model 3 (M3) in the presence (filled brown triangle for M2; purple asterisk for M3) and absence (empty blue triangle for M2; pink × for M3) of inhibitory ligands (HLA-C). The parameters for the simulation are set to the best-fit values. The NKG2D concentrations are averaged over 200 different configurations. KIR2DL2 inhibition abrogates centripetal movements of NKG2D receptor clusters for Model 2 but not for Model 3.
(TIF)

**S7 Fig. Effect of HLA-C spatial patterning on target cells on signal integration in NK cells.** We co-localized KIR-2DL2-HLA-C complexes in a ring pattern at t = 0 in our simulations as shown in Fig 6A. NKG2D, ULBP3 and other parameters are set as described in the section pertaining to Fig 6. The figure shows variation of total number of pVav1 at t = 1 min with increasing HLA-C concentrations for Model 2 (empty blue triangle) and Model 3 (purple asterisks). The decrease in pVav1 is higher in Model 2 compared to Model 3. The pVav1 values shown were obtained by averaging over 50 configurations for each HLA-C dose.
(TIF)

**S8 Fig. Model prediction for NKG2D spatial clustering at later times are in agreement with live cell confocal microscopy reported in Ref. [7].** We extracted two-dimensional fluorescence intensity of NKG2D-GFP in single NKL published in Ref. [7] following an image extraction method described in the Materials and Methods section. The NKLs in Ref. [7] were stimulated by MICA ligands on target Daudi/MICA cells and NKG2D-GFP molecules were imaged using confocal microscopy. We chose a region of interest in the extracted image and coarse-grained it to match the minimum length scale (0.5 μm) of our simulation. We used the confocal data at t = 0 (A; top) to create the initial configuration of NKG2D receptors in the model (Model 2). The rest of the parameters are initialized as described in the Materials and Methods sections. We simulated the initial configuration in our model (Model 2) using the best fit parameters (Table 3) we obtained for Model 2 using the TIRF imaging data in Ref. [6].

The spatial organization of NKG2D in the simulation (top panels in (A)-(C)) and experiments (bottom panels in (A)-(C)) at t = 30s and t = 70s were compared using the two-point correlation function (D-E). The two-point correlation functions in the simulations are averaged over an ensemble of 100 configurations. The values for the correlation function for individual simulation trajectories are shown in grey.
(TIF)

**S9 Fig. Approximating cytosolic diffusion by homogenization of molecules at discrete times has negligible effect on pVav1 kinetics for Model 1.** Shows the total number of pVav1 with time in Model 1 when diffusion of cytosolic molecules is introduced explicitly in the simulations (black, empty circle), which is compared against our simulations approximating explicit diffusion with homogenization of those molecules at discrete times (magenta, asterisk). The pVav1 concentrations are averaged over 200 different configurations. The parameters for the simulation are set to the best-fit values from PSO.
(TIF)

**S10 Fig. Effect of intrinsic noise fluctuations in C(r,t).** Shows probability distribution function (pdf) of $C(r,t)/C(0,t)$ at t = 1min for (**A**) $r = 1\mu m$ and (**B**) $r = 4\mu m$ for Model 2. The parameters for the simulation are set at the best-fit value from our PSO. The pdf is calculated for an ensemble of 200 configurations. $\mu$ and $\sigma$ represent the mean and standard deviation of the pdf, respectively. $c^*$ denotes the values of the $C(r,t)/C(0,t)$ at t = 1min at the r values shown in (A) and (B) for the best-fit NKG2D configuration obtained in the PSO.
(TIF)

**S11 Fig. Effect of the intrinsic noise fluctuations in the cost function $C_{cost}(\theta_{min}; \eta, n_0)$ for the best -fit parameter value $\theta_{min}$ for an initial state $n_0$.** $\eta$ denotes the random variables associated with intrinsic noise fluctuations. The pdf is calculated for an ensemble of 400 configurations. $\mu$ and $\sigma$ denotes the mean and standard deviation for $C_{cost}(\theta_{min}; \eta, n_0)$ for the corresponding model. $C_{opt}$ represents the optimum (minimum) cost function $C_{cost}(\theta_{min}; \eta = \eta_{pso\_otim}; n_0)$ estimated by PSO for each model. The uncertainty in our estimated parameters $\theta_{min}$ is estimated to lie within the interval $0 \leq C_{cost} \leq C_{thres}$, where $C_{thres}$ is obtained as $C_{thres} = C_{opt} + 2\sigma$.
(TIF)

**S12 Fig. Decision Graph to compute clusters according to Density Peak Clustering algorithm (Ref. [92]).** The points having relatively large local density ($\rho$) and high value of $\delta$ are considered to be the cluster centers defined in Ref. [92]. For both, Model 1 (A) and Model 2 (B), we observed only one such cluster center to exist implying the presence of a single minimum cost function in the parameter range explored by the PSO. The decision graph was computed for parameters $\theta$ in the solution space having cost function, $C_{cost}$ such that $0 \leq C_{cost} \leq C_{thres}$, where $C_{thres} = 0.29$ and $0.35$ for Model 1 and Model 2, respectively. The parameter $d_c$ in the Density Peak Clustering algorithm in Ref. [92] is calculated such that the average number of neighbours is 2% of the total number of points in the data set.
(TIF)

**S13 Fig. Image extraction.** (A) Region of interest extracted from S4 Fig at t = 1min in Ref. [6]. (B) Coarse-grained image of region of interest in (A) to obtain the minimum scale length resolution ($\sim 0.5 \mu m$) in our simulations. The color bar shows the extracted intensities of Dap10-mCherry from the TIRF experiments in Ref. [6].
(TIF)

**S14 Fig. Effect on pVav1 kinetics due to change in parameters $k_{on\{Dap10:SFK\}}$, and $k_{off\{Dap10: SFK\}}$.** Shows the total number of pVav1 with time for Model 2 for the values of these rates

estimated in our PSO (blue triangle), estimations from the literature for SFK:NKG2D (orange circle), and Lck:CD3ζ (green diamond). The estimated values are shown in S1 Text. The pVav1 concentrations are averaged over 50 different configurations and show no appreciable differences for the three sets of rates that were used.
(TIF)

**S15 Fig. Effect changing the binding rate ($k_{on}$) of the catalytic domain of SHP1 to pVav1 on pVav1 kinetics.** Shows pVav1 kinetics in the presence of activating ULPB3 and inhibitory HLA-C ligands. The $k_{on}$ value was decreased 10× (cyan circles) and 150× (purple diamonds) from the value (6.11 $\mu M^{-1} s^{-1}$) used in Model 2 (brown triangle). The pVav1 kinetics in with activating ULBP3 and in the absence of inhibitory ligands is shown as a reference (blue empty triangle). KIR2DL2 is distributed in the simulation box following the data extracted from TIRF imaging at 1 min 44 sec in Fig 4 of Ref. [6]. The pVav1 concentrations are averaged over 50 different configurations.
(TIF)

**S16 Fig. Modeling cytotoxic killing assay in presence of treated and untreated target cells.** (**A**) Shows normalized distribution of NKG2D ligands in the treated and untreated target cells extracted from the Fig. 2d in Ref. [63] using a graphing software. (**B**) Distribution of NKG2D ligands when a sample of 1000 target cells (filled bars) are drawn from the normalized distribution for the case of treated target cells shown in dark-green filled squares. (**C**) Percentage lysis for the population of 1000 treated or untreated target cells. (**D**) Percentage lysis of target cells corresponding to data from Fig. 2d in Ref. [63].
(TIF)

**S1 Text. Estimation of rate for binding and unbinding of catalytic domain of SFK interacting with tyrosine residues in (Dap10 –NKG2D) based on the estimated values of $k_{off}$, $K_m$, $k_{cat}$ from literature.**
(DOCX)

**S2 Text. Rates for de-phosphorylation of pVav1 by KIR2DL2 bound SHP1.**
(DOCX)

**S3 Text. Modeling cytotoxic killing assay.**
(DOCX)

**S1 Movie. Shows the kinetics of spatial clustering of NKG2D in presence of its cognate ligand ULBP3 in the x-y plane in the simulation box for Model 1.** The parameters for the simulation are set at the best fit value from our PSO.
(MP4)

**S2 Movie. Shows the spatial clustering of NKG2D in presence of its cognate ligand ULBP3 in the x-y plane in the simulation box for Model 2.** The parameters for the simulation are set at the best fit value from our PSO.
(MP4)

**S3 Movie. Shows the decrease in clustering of NKG2D upon inhibition by KIR2DL2 in the x-y plane in the simulation box for Model 2.** The parameters for the simulation are set at the best fit value from our PSO. Kinetic rates describing inhibition are provided in Table 1.
(MP4)

**S4 Movie. Shows clustering of NKG2D in the presence of inhibitory ligands in the absence of the pVav1 induced spatial positive feedback.** The video shows clustering behavior of

NKG2D in the x-y plane in the simulation box for Model 3. The parameters for the simulation are set at the best fit value from our PSO. Kinetic rates describing inhibition are provided in Table 1.
(MP4)

**S5 Movie. Movement of microclusters of different sizes.**
(MP4)

**S1 Data. Excel spreadsheet containing, in separate sheets, the underlying numerical data for panels Figs 2D, 3C, 3D, 4A, 4B, 4C, 4D, 5B, 5C, 6C and 6D.**
(XLSX)

**S2 Data. Data for Fig 2A.** Each value in the .vtk data file corresponds to the intensity of NKG2D-DAP10-mCherry, extracted from TIRF experiments in Ref. [6] at t = 1min post stimulation by ULPB3, at a x-y grid location in the simulation box. "DIMENSIONS" and "POINT DATA" in the header of the data file represent the dimensions of the simulation box and total number of chambers in the simulation box, respectively. The image is generated using Para-View-5.7.0 which takes .vtk files as inputs.
(VTK)

**S3 Data. Data for Fig 2B.** Each value in the .vtk data file corresponds to the number of NKG2D in each chamber of the simulation box for our Model 1 at t = 1 min post NKG2D stimulation. "DIMENSIONS" and "POINT DATA" in the header of data file represent the dimensions of the simulation box and total number of chambers in the simulation box, respectively. The image is generated using ParaView-5.7.0 which takes .vtk files as inputs.
(VTK)

**S4 Data. Data for Fig 2C.** Each value in the .vtk data file corresponds to the number of NKG2D in each chamber of the simulation box for our Model 2 at t = 1 min, post NKG2D stimulation. "DIMENSIONS" and "POINT DATA" in the header of data file represent the dimensions of the simulation box and total number of chambers in the simulation box, respectively. The image is generated using ParaView-5.7.0 which takes .vtk files as inputs.
(VTK)

**S5 Data. Data for Fig 3A (top).** Each value in the .vtk data file corresponds to the intensity of NKG2D-DAP10-mCherry, extracted from TIRF experiments in Ref. [6] at t = 2min, post stimulation by ULPB3, at a x-y grid location in the simulation box. "DIMENSIONS" and "POINT DATA" in the header of the data file represent the dimensions of the simulation box and total number of chambers in the simulation box, respectively. The image is generated using Para-View-5.7.0 which takes .vtk files as inputs.
(VTK)

**S6 Data. Data for Fig 3A (middle).** Each value in the .vtk data file corresponds to the number of NKG2D in each chamber of the simulation box for our Model 1 at t = 2 min, post NKG2D stimulation. "DIMENSIONS" and "POINT DATA" in the header of data file represent the dimensions of the simulation box and total number of chambers in the simulation box, respectively. The image is generated using ParaView-5.7.0 which takes .vtk files as inputs.
(VTK)

**S7 Data. Data for Fig 3A (bottom).** Each value in the .vtk data file corresponds to the number of NKG2D in each chamber of the simulation box for our Model 2 at t = 2 min, post NKG2D stimulation. "DIMENSIONS" and "POINT DATA" in the header of data file represent the dimensions of the simulation box and total number of chambers in the simulation box,

respectively. The image is generated using ParaView-5.7.0 which takes .vtk files as inputs. (VTK)

**S8 Data. Data for Fig 3B (top).** Each value in the .vtk data file corresponds to the intensity of NKG2D-DAP10-mCherry, extracted from TIRF experiments in Ref. [6] at t = 3min, post stimulation by ULPB3, at a x-y grid location in the simulation box. "DIMENSIONS" and "POINT DATA" in the header of the data file represent the dimensions of the simulation box and total number of chambers in the simulation box, respectively. The image is generated using ParaView-5.7.0 which takes .vtk files as inputs. (VTK)

**S9 Data. Data for Fig 3B (middle).** Each value in the .vtk data file corresponds to the number of NKG2D in each chamber of the simulation box for our Model 1 at t = 3 min, post NKG2D stimulation. "DIMENSIONS" and "POINT DATA" in the header of data file represent the dimensions of the simulation box and total number of chambers in the simulation box, respectively. The image is generated using ParaView-5.7.0 which takes .vtk files as inputs. (VTK)

**S10 Data. Data for Fig 3B (bottom).** Each value in the .vtk data file gives the number of NKG2D in each chamber of the simulation box for our Model 2 at t = 3 min, post NKG2D stimulation. "DIMENSIONS" and "POINT DATA" in the header of data file represent the dimensions of the simulation box and total number of chambers in the simulation box, respectively. The image is generated using ParaView-5.7.0 which takes .vtk files as inputs. (VTK)

**S11 Data. Data for Fig 5A.** Each value in the .vtk data file gives the number of NKG2D in each chamber of the simulation box for our Model 2 in the presence of both activating ULBP3 and inhibitory ligands at t = 1 min. "DIMENSIONS" and "POINT DATA" in the header of data file represent the dimensions of the simulation box and total number of chambers in the simulation box, respectively. The image is generated using ParaView-5.7.0 which takes .vtk files as inputs. (VTK)

**S12 Data. Data for Fig 6B.** Each value in the .vtk data file gives the number of NKG2D in each chamber of the simulation box for our Model 3 in the presence of both activating ULBP3 and inhibitory ligands at t = 1 min. "DIMENSIONS" and "POINT DATA" in the header of data file represent the dimensions of the simulation box and total number of chambers in the simulation box, respectively. The image is generated using ParaView-5.7.0 which takes .vtk files as inputs. (VTK)

## Acknowledgments

We thank Salim I. Khakoo and C. Jayaprakash for discussions. RKG thanks Jonathan Brown, Darren Wethington, Indrani Nayak, and Ali Snedden for help with simulations.

## Author Contributions

**Conceptualization:** Rajdeep Kaur Grewal, Jayajit Das.

**Formal analysis:** Rajdeep Kaur Grewal, Jayajit Das.

**Funding acquisition:** Jayajit Das.

**Investigation:** Rajdeep Kaur Grewal.

**Methodology:** Rajdeep Kaur Grewal, Jayajit Das.

**Project administration:** Jayajit Das.

**Resources:** Rajdeep Kaur Grewal.

**Software:** Rajdeep Kaur Grewal.

**Supervision:** Jayajit Das.

**Visualization:** Rajdeep Kaur Grewal, Jayajit Das.

**Writing – original draft:** Rajdeep Kaur Grewal, Jayajit Das.

**Writing – review & editing:** Rajdeep Kaur Grewal, Jayajit Das.

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
