## [Decision Letter · Decision Letter 0]

6 Jan 2022

Dear Prof. Das,

Thank you very much for submitting your manuscript "Spatially Resolved In Silico Modeling of NKG2D Signaling Kinetics Reveals Key role of NKG2D and Vav1 Co-clustering in Generating Natural Killer cell Activation" for consideration at PLOS Computational Biology.

As with all papers reviewed by the journal, your manuscript was reviewed by members of the editorial board and by several independent reviewers. In light of the reviews (below this email), we would like to invite the resubmission of a significantly-revised version that takes into account the reviewers' comments.

Both reviewers expressed concerns about the results. In particular, Rev.2 has serious concerns

about validation of the model of biochemical signaling reactions as well as fitting the model with

the data and possible overfitting problem. Rev.1 is also concerned about overfitting as well as

about parameter values estimation. 

We cannot make any decision about publication until we have seen the revised manuscript and your response to the reviewers' comments. Your revised manuscript is also likely to be sent to reviewers for further evaluation.

Sincerely,

Mark Alber, Ph.D.

Deputy Editor

PLOS Computational Biology

Mark Alber

Deputy Editor

PLOS Computational Biology

Reviewer's Responses to Questions

**Comments to the Authors:**

Reviewer #1: This manuscript describes a spatial agent-based model for investigating the interaction between signalling and spatial organisation of receptors and adaptor proteins in the immunological synapses of NK cells. This is an important question for understanding the function of NK cells, as well as T cells that also form immunological synapses and integrate information between activating and inhibitory receptors. The authors show that their model captures key features of previous experimental imaging data of NKG2D clustering and dynamics in NK cell synapses and that co-clustering between Vav1 and NKG2D is more important for robust signalling than rebinding of NKG2D to its ligand. In their modelling Grewal and Das show that NKG2D translocation to the centre of the synapse could initiate a positive feedback of Vav1 phoshorylation, if translocation is movement is dependent on pVav1, and that this allows for efficient inhibition by KIR2DL1.

Overall the work presents an interesting hypothesis that is supported by previous microscopy studies. I have some questions surrounding parameter estimation, and suggest some extra discussion of limitations, which are outlined below:

Suggest softening the claim in the title from "…Reveals role…" to "…Suggests role…" or "…is consistent with a role…". The authors show their model is consistent with previous work, but to establish that the mechanism they propose exists in the biology, some experimental validation of novel predictions made by the model would be required.

Table 1, Rule 2: Are these binding rates the estimate for the catalytic domain of the SFK interacting with the unphosphorylated DAP10 substrate (ie their ratio would be the Michealis-Menton constant KM)? If so why are the kinetics of SH2 domain for pY used to estimate this? They are different domains and completely different interactions. Others have estimated these values (doi.org/10.1007/s12195-016-0438-7).

Table 1, Rule 13: Why is ITIM-bound SHP1:Vav1 binding kon set to 10x the Vav1:pDAP10 kon rate and set to have the same koff? Why was this parameter not estimated by PSO, given it depends on local concentrations and steric availability of substrates and enzymes tethered to receptors and is thus very difficult to measure and even estimate rationally (see DOI: 10.1126/sciadv.1601692 and DOI: 10.1016/j.bpj.2019.08.023).

Table 1, Rule 14: Catalytic rate on pNPP is a very poor indicator of catalytic rates on specific pY targets, which can have large variability depending on flanking sequences. For measurements of SHP1 catalytic rates on phosphotyrosine substrate see doi.org/10.1016/j.bpj.2021.03.019 and others.

Around line 489, a little more discussion about the limitation of not modelling cellular spreading and retraction, both of which are related to activating and inhibitory receptor signalling (eg in doi.org/10.1083/jcb.201009135), and related to Vav1 activity. It would be useful for placing this work in this context. Splitting the limitations section into paragraphs would also make it more readable.

Reviewer #2: In this paper, Grewal and Das create an agent based model of NKG2D and it’s signaling kinetics following binding with cognate ligands. As NK cells are activated and inactivated by a diverse set of signaling receptors, this work provides in silico insight into these signaling transduction pathways. Further, they add value to the field by tying capturing spatial signaling dynamics into their models.

There were some concerns:

1) I would have liked to see more validation on their model on other signaling pathways, especially their modeling of biochemical signaling reactions. For example, how well would it predict findings from this paper on NKG2D: doi.org/10.1038/nature03884.

2) Along the same lines, would the modeling have to be fitted each time for use in specific signaling systems or would their system be useful to describe microcluster formation in general? For example, the authors cite Steblyanko et. al. as a limitation of their work, however, would the in silico modeling be predictive of these results? If not, I would share Reviewer 1’s concerns about overfitting. Also, it would limit the impact of this type of in silico modeling.

3) The other concern related to overfitting, their biochemical signaling processes are limited to a handful of papers. The models would rely heavily on the accuracy of the original paper findings. Can the authors validate their parameters across different publications/published results? Without external validation of the modeling, it’s difficult to evaluate the significance of the findings.

4) During the discussion of microclusters, please cite these papers: doi.org/10.3389/fimmu.2012.00421, doi: 10.1038/nri2381.

**Have the authors made all data and (if applicable) computational code underlying the findings in their manuscript fully available?**

Reviewer #1: Yes

Reviewer #2: Yes

PLOS authors have the option to publish the peer review history of their article (what does this mean?). If published, this will include your full peer review and any attached files.

Reviewer #1: No

Reviewer #2: No
---

## [Decision Letter · Decision Letter 1]

18 Apr 2022

Dear Prof. Das,

We are pleased to inform you that your manuscript 'Spatially Resolved In Silico Modeling of NKG2D Signaling Kinetics Suggests a Key role of NKG2D and Vav1 Co-clustering in Generating Natural Killer cell Activation' has been provisionally accepted for publication in PLOS Computational Biology.

Best regards,

Mark Alber, Ph.D.

Deputy Editor

PLOS Computational Biology

Mark Alber

Deputy Editor

PLOS Computational Biology

Reviewer's Responses to Questions

**Comments to the Authors:**

Reviewer #1: Thank you for addressing my comments and making changes to your manuscript. I have no further concerns.

Reviewer #2: All my concerns have been addressed with the new experiments performed to demonstrate the utility of their model in other settings.

**Have the authors made all data and (if applicable) computational code underlying the findings in their manuscript fully available?**

Reviewer #1: Yes

Reviewer #2: None

PLOS authors have the option to publish the peer review history of their article (what does this mean?). If published, this will include your full peer review and any attached files.

Reviewer #1: No

Reviewer #2: No

---

## [Editor Report · Acceptance letter]

11 May 2022

PCOMPBIOL-D-21-02044R1 

Spatially Resolved In Silico Modeling of NKG2D Signaling Kinetics Suggests a Key role of NKG2D and Vav1 Co-clustering in Generating Natural Killer cell Activation

Dear Dr Das,

I am pleased to inform you that your manuscript has been formally accepted for publication in PLOS Computational Biology. Your manuscript is now with our production department and you will be notified of the publication date in due course.

With kind regards,

Livia Horvath
